
**Performance Evaluation for Retrieving Aerosol Optical Depth from**
**Directional Polarimetric Camera (DPC) based on GRASP Algorithm**
Shikuan Jin [1], Yingying Ma [1, 2, *], Cheng Chen [4], Oleg Dubovik [5], Jin Hong [6], Boming Liu [1], Wei
Gong [2, 3]
[1] State Key Laboratory of Information Engineering in Surveying, Mapping and Remote Sensing,
Wuhan University, China.
[2] Collaborative Innovation Center for Geospatial Technology, Wuhan 430079, China.
[3] School of Electronic Information, Wuhan University, China.
[4] GRASP-SAS, Remote Sensing Developments, Cite Scientifique, University of Lille, 59655
Villeneuve d'Ascq, France.
[5] Univ. Lille, CNRS, UMR 8518 - LOA - Laboratoire d'Optique Atmosphérique, Lille, France.
[6] Anhui Institute of Optics and Fine Mechanics, Chinese Academy of Sciences, Hefei 230031,
China.
Corresponding Author: Yingying Ma (yym863@whu.edu.cn)
**Abstract**
Aerosol spatial distribution obtained from the satellite sensor is a critical point to understand
regional aerosol environment, anthropogenic aerosol emissions, and global climate change. In this
study, the performance of aerosol optical depth (AOD) retrieval from the Directional Polarimetric
Camera (DPC)/GaoFen-5 by using the Generalized Retrieval of Atmosphere and Surface Properties
(GRASP) algorithm was evaluated on a global basis for the first time. The results showed that the
DPC GRASP/Model scheme, which used several aerosol-type mixings, achieved good performance.
By compared with AERONET observations, the correlation coefficient (R), normalized mean square
error, and Expect Error (EE%) were 0.8982, 0.1008, and 83.16%, respectively. The scattering angle,
number of averaged pixels in retrieval units, and radiative and polarized fitting residuals showed
impacts on the results of AOD retrieval in the DPC GRASP/Model. From the most of AERONET
sites, the R and EE% were larger than ~0.9 and ~80%. Compared with MODIS products, the spatial
and temporal variations of AOD could be caught by the DPC observations with the GRASP/Model,
and compared with the MODIS Dark Target algorithm, the DPC GRASP/Model AOD also showed
a good performance. The above findings validated the ability of DPC sensor to monitor aerosols. It
would contribute to the development of aerosol parameter retrieval from multi-angular polarized
sensors in the future.

**Key Words:** GRASP/Model, Aerosol Optical Depth, Directional Polarimetric Camera, GaoFen-5,
Aerosol Parameter Retrieval



## 1. Introduction

Aerosol is one of the most important components in the atmosphere. They influence the global radiation budget balance and climate directly by scattering and absorbing incoming solar radiation and indirectly by changing cloud microphysical properties (Albrecht 1989; D'Almeida et al. 1991; Rosenfeld et al. 2008). Due to the different emission sources and relatively transitory lifecycle in the atmosphere, aerosol particles show large spatiotemporal variability, and it is difficult to describe uniformly at a global scale (Eck et al. 2010; Jin et al. 2019; Ma et al. 2021). This property can further affect the atmospheric motion, hydrological cycle, and probably contribute regional extreme weather events (Nakajima et al. 2007; Guo et al. 2016; Li et al. 2016; Shi et al. 2021). Therefore, the development of aerosol measurement technologies has been a topic received widely attention in recent decades.

Satellite observation is the mainly approach to monitor and quantify aerosol distributions at a global scale (Kaufman et al. 1997). Traditional Satellite technology relies on unique channel design and prior assumptions about the properties of the surface and atmosphere, because the prerequisite for successful retrieval of aerosol is that the aerosol signal should be isolated from a total mixture of information received by satellite, which includes the combined effect from molecule, aerosol, cloud, and the underlying surface (Lenoble et al. 2013). For instance, the appropriate spatial resolution helps to observe aerosol through clear holes in otherwise cloudy skies (Jin et al. 2021). The choice of spectral channel and bandwidth can avoid impact by gas absorption, if they are in narrow spectral bands of atmosphere window regions. In addition, more importantly, the spectral channel should be set in a carefully selected band to avoid introducing uncertainty from underlying surface features in the meantime, such as vegetation, bright desert, and ocean color (McCormick et al. 1979; Rao et al. 1989; Hsu et al. 2004). Based on these principles, a series of aerosol products from different sensors has been released, and they greatly promote the developments of studies in aerosol-related fields, including aerosol climate effect, interaction of aerosol and cloud, air quality and public health, and global climate modeling (Tegen and Lacis 1996; Sayer et al. 2013; Gao et al. 2017; Zhang et al. 2021).

With the progress of satellite technology, sensors with broader spectral range, multiple angles, and polarization observations have also been applied to aerosol observations. The POLDER-3 is the third sensor in the POLarization and Directionality of the Earth's Reflectance series, carried on the Polarization and Anisotropy of Reflectances for Atmospheric Science coupled with Observations from a Lidar (PARASOL), which was launched on December 18, 2004, as part of the A-Train (Tanre et al. 2011). This instrument views (±51° along track and ±43° across track) Earth from ~13 different angles by using a set of wide-field telecentric optics and a rotating filter wheel in nine spectral channels from 443 to 1020 nm (Deschamps et al. 1994). Among them, three channels in 490, 670, and 865 nm have polarization observation capabilities. The POLDER-3 provides the longest multi-angle polarimetric observation record of the Earth-atmosphere system in space to date and the PARASOL mission was terminated in December 2013 due to limited on-board fuel budget. The Directional Polarimetric Camera (DPC) is the first Chinese multi-angle polarized earth observation satellite sensor, onboard the fifth satellite (GaoFen-5) of the Chinese High-resolution Earth Observation Program (Li et al. 2018). It was launched successfully on May 9, 2018, with the purposes of measuring aerosol parameters and providing information for the assessment of urban air pollution. The design of DPC is similar to the POLDER-3. It is equipped five non-polarized



bands at 443, 565, 763,765, and 910 nm and three polarized bands at 490, 670, and 865 nm, with
relatively higher spatial resolution of 3.3 km, that can observe Earth from ~9 different angles.
Therefore, the DPC occupies an important position in the development of polarization instruments
in China, and is expected to provide beneficial information for atmospheric aerosol monitoring and
satellite payload research.
87       The multi-angular polarized sensor can provide much more observations for the same pixel in
aerosol parameter retrieval. Compared to traditional spectral measurement, the multi-angle can help
constrain bidirectional reflections function, reducing uncertainty from the surface (Diner et al. 1998),
while the polarized signal is mainly from atmospheric aerosol and sensitive to particle microphysical
properties (Mishchenko and Travis 1997). Generally, the polarized signal can be considered as an
independent source of information. A well-known advantage is that the polarized light from the
surface is accounts for a small part of the total polarized light compared with that from the particles
and shows a feature of almost wavelength independence. In the algorithms for POLDER, the
polarized signals at 670 and 865 nm are used for deriving the best aerosol model over the ocean and
retrieving Aerosol Optical Depth (AOD) over land, due to the sensitivity to fine particles (Nadal
and Bréon 1999; Deuzé et al. 2001; Kacenelenbogen et al. 2006; Ge et al. 2020). In addition, the
existence of the cloudbow effect in polarized signal can also be used to recognize cloud mask and
detect cloud structure (Breon and Goloub 1998; Breon and Colzy 1999; Li et al. 2021).
However, the algorithms that retrieve aerosol parameters from only one or two polarized
channels are still difficult to obtain complex aerosol optical and microphysical parameters, such as
aerosol size distribution and absorbing and scattering properties. To solve this problem, the
Generalized Retrieval of Atmosphere and Surface Properties (GRASP) algorithm is developed,
which provides a novel statistical optimized strategy that allows all aerosol-related measurement
data from multi-angular polarized sensors to participate in the retrieval (Dubovik et al. 2014). It
points out that the measured redundancy provided by multi-angular polarized sensor is considered
to be positive and useful, especially when the observations are larger than the unknowns (Dubovik
et al. 2011). At present, the GRASP algorithm has been successfully applied to a variety of sensors
to retrieve complex aerosol parameters, including POLDER, lidar, and sun photometer (Li et al.
2019; Chen et al. 2020; Lopatin et al. 2021). In this study, we retrieved AOD from DPC observations
by using GRASP algorithm and evaluated possible error influencing factors. At the same time, by
comparing MODIS and AERONET observations, the aerosol monitoring performance of DPC were
verified in different space and time scales. This will partially lay the foundation for the retrieval of
aerosol parameters from multi-angular polarized sensors in the future of China.

## 2. Satellite and Ground-based Data

### 2.1 DPC Data

The DPC is a multi-angular polarized sensor carried on the GF-5 satellite, which was launched
in May 9, 2018. This sensor completes a scan of entire Earth's surface about every two days at a
sun-synchronous orbit and provides a swath of 1850 km with a spatial resolution of 3.3 km (Li et
al. 2018). The DPC contains eight bands from 443 to 910 nm with a bandwidth of 10-40 nm that
can observe earth from ~9 different angles in a local time of ~13:30 PM. Except for water vapor
band (910 nm) and pressure bands (Oxygen A band: 763 and 765 nm), other five bands (443, 490,



565, 670, and 865 nm) are designed for observing aerosol (Li et al. 2018). The polarimetric
capability at 490, 670, and 865 nm is realized by a polarized filter wheel (0°, 60°, and 120°) and a
step motor (Hagolle et al. 1999). The laboratory calibration uncertainties are relatively 5% for
normalized radiation and absolutely 0.02 for Degree Of Linear Polarization (DOLP) (Li et al. 2021).
An in-flight calibration study showed that the radiometric calibration error increased to ~9% at 865
nm and the polarimetric calibration error increase to ~0.04 at 490 and 670 nm after launch, by
respectively applying Rayleigh and glint scenes over ocean (Qie et al. 2021). While, degradation of
instrument performance over time may result in higher negative radiometric shift (Zhu et al. 2022).
Thus, additional correction coefficients were also applied in this study to correct the image of the
DPC observations from March to April, 2020. For preparing to retrieve AOD, the processing of
DPC data is described in Section 3.2 in detail.
### 2.2 MODIS aerosol products
The Moderate-resolution Imaging Spectroradiometer (MODIS) has been in service for over
two decades, providing valuable data for the earth's observations. The MODIS Level 2 C6.1 aerosol
product (MxD04) is generated by using Dark Target (DT) algorithm and Deep Blue (DB) algorithm
(Hsu et al. 2013; Levy et al. 2013). It provides multi-wavelength AOD data from each individual
image with spatial resolutions of 3 km and 10 km. While, the MODIS Level 2 C6 aerosol product
(MCD19A2) considers temporal and spatial correlation of aerosols, calculating aerosol parameters
by using the Multi-Angle Implementation of Atmospheric Correction (MAIAC) algorithm from the
continuous scenes of two satellites (Terra and Aqua), with a spatial resolution of 1 km (Lyapustin et
al. 2018). Compared to global coverage of DT algorithm, the DB algorithm is only applied over
land, and the MAIAC algorithm is used over land and part of the surrounding ocean. These MODIS
aerosol products have been rigorously tested and verified, and are widely used in aerosol-related
studies (Sayer et al. 2014; Che et al. 2019; Zhdanova et al. 2020). Only MODIS data with the highest
quality were used in this study.
### 2.3 AERONET observations
The AErosol RObotic NETwork (AEROENT) is a federation of ground-based remote sensing
aerosol networks, established and expanded by various institutions from different countries (Holben
et al. 1998). It has contributed continuous and long-term aerosol optical, microphysical, and
radiative properties for more than 25 years in major ecosystems and human activity areas around
the world. The AOD data used for validation were acquired from Level 1.5 and Level 2.0
AERONET products, which have been cloud-screened and quality controlled. The uncertainties of
AOD are less than 0.02 (Eck et al. 1999). In order to match the AERONET data to the satellite
observations, a common approach is followed to averages satellite data within ±30 min and a circle
of 0.25° (~25 km) radius centered at the selected site (Sayer et al. 2013). The relationship between
multi-wavelength AOD proposed by Ångstrom (1964) was applied to calculate the AOD at
corresponding wavelength of satellite bands from AERONET data.
## 3. Methods
### 3.1 Introduction of GRASP algorithm
GRASP is an open-source software package (https://www.grasp-open.com/) for calculating
and retrieving various optical and microphysical properties of aerosol and surface from observations
of different remote sensing instruments, such as satellite, lidar, radiometer, and radiosonde (Dubovik
et al. 2021). It was originally designed to improve and solve the problem of aerosol retrieval under
high surface reflectance conditions from the PARASOL observations (Dubovik et al. 2014), while
now has become a scientifically rigorous and versatile algorithm based on generalization principles
that works with diverse remote sensing applications in the community after continuous development
(Dubovik et al. 2021). The GRASP algorithm contains two pivotal and independent modules. One
is used to calculate the scattering, absorbing, and extinction of light between different media from
the physical level, simulating theoretical observational radiation signal, called "Forward Model". It
allows define various complex aerosol (size distribution, refractive index, and sphere fraction, etc.)
and surface properties (Bidirectional Reflectance/ Polarization Distribution Function, BR/PDF, etc.)
in the construction of model. Therefore, this makes it possible to transform from optical observations
to aerosol microphysical properties and estimate the surface parameters in the meantime (Dubovik
et al. 2011). The other module can be thought of as general mathematical operations without any
particularly physical nature, called "Numerical Inversion". It follows the statistically optimized
strategy to fit observations under the fundamental frameworks of the Maximum Likelihood Method
and multi-term Least Square Method (Dubovik and King 2000). By introducing Lagrange multiplier
method, the GRASP also realizes multiple-pixel retrieval, which constrains the variability of aerosol
and surface optical properties in fitting process by an extra prior knowledge. Due to the consideration
of the surrounding pixel information, the multi-pixel retrieval is more stable, and more importantly,
it can make up for the lack of aerosol reflection information in some cases, such as conditions that
the signal from aerosol is much less than that from the surface (Dubovik et al. 2011). Based on the
above advantages, the GRASP supports input measurements/parameters from different sources and
levels, such as normalized and polarized radiance, vertical extinction and backscatter profile, and
optical depth. This avoids that the traditional look-up table-based methods are difficult to apply to
each other, due to the limitations of different sensor channel and characteristic.

### 3.2 Pre-processing of DPC Data

In order to partially offset the signal attenuation due to possible instrument aging, before the
pre-processing and retrieval, the radiance signals from the DPC were transferred and corrected to
normalized radiative and polarized reflectance at top of the atmosphere.

$$[I_N, Q_N, U_N]^T = \pi \cdot [I, Q, U]^T / [E_0 \cdot A'_k(\theta_0) \cdot P'_k(\theta)] \tag{1}$$

where, the $[I, Q, U]^T$ are represent the radiative and polarized radiances, received by the DPC, in
the form of the first three parts of the Stokes vector. The $A'_k(\theta_0)$ and $P'_k(\theta)$ are the two additional
correction coefficients. For $I$, they are applied following the results of Zhu et al. (2022), which are
depended on the view zenith angle ($\theta$) and calculated based on Rayleigh scenes over sea surface.
For polarimetric signals, the additional correction coefficients can be referred to Qie et al. (2021).
The $E_0$ is the standard solar radiation flux and the $[I_N, Q_N, U_N]^T$ are the corrected normalized
signals at top of the atmosphere of DPC.
In successful AOD retrieval, one of the key processes is to screen appropriate pixels. Cloud
pixel is the main factor impacting aerosol retrieval, because they will block the signal from aerosol
due to high reflectance, large coverage, and relatively high vertical position. Even very thin cirrus
clouds and missed cloud edges can cause an obviously positive error of ~13% in visible channel



(Koren et al. 2007). To remove cloud pixels in DPC images, we used several universal methods by
considering cloud-sensitive characteristics in radiative and polarized bands:
1) The first step is to filter the image with a 3×3 sliding window in blue (490 nm) and red (670
nm) bands for land and sea surfaces, respectively (Remer et al. 2012). If the standard deviation of a
window is greater than 0.0025, then the center pixel will be marked as a cloud pixel and removed
(Martins et al. 2002). This method was initially applied to the MODIS image by considering the
spatial variability of aerosol and cloud pixels. In addition, a threshold of > 0.4 in the green (565 nm)
band is also used to detect cloud pixels after the filter process, in accordance with the DT algorithm.
This threshold is to exclude very uniformly distributed cloud pixels in the central area of thick clouds,
and some snow pixels and glint area will also be excluded at the same time.
2) In second step, a whiteness test was applied by using reflectance in visible bands. It uses the
characteristic that clouds are white in the visible band, considering that pixel with the absolute value
of average relative deviations greater than 0.7 is cloud. In the absence of infrared and thermal
infrared information, it can supplementally remove any pixels that have flat reflectance, similar to
some operators using reflectance ratio to detect clouds. This method was proposed by Gomez-Chova
et al. (2007) for Medium Resolution Imaging Spectrometer (MERIS) multispectral image, and it
has also been considered in the well-known Fmask algorithm.
3) The third step used polarized bands to remove cloud pixels, following a fact that cloud drops
can show a relatively strong polarized reflectance by multiple scattering (cloudbow effect) under a
specify observation geometry. This feature has been used to generate cloud mask product for both
POLDER and DPC sensors (Breon and Colzy 1999; Li et al. 2021). When the scattering angle (SCA)
is between 127° and 157°, pixels with corrected polarized radiation at 865 nm larger than 0.03 and
0.05 for ocean and surface, respectively, are defined as cloud (Li et al. 2021). The relatively large
SCA range is for a strict screening, given that the main peak of the polarized reflectance by cloud
water droplets is ~142° (Goloub and Deuze 1994). In addition, any obvious noise is also removed
in this step, such as the case of DOLP > 1.

### 3.3 Construction of Multi-pixel Retrieval Unit

Next, we will explain the necessary operations and settings of parameters to apply the GRASP
algorithm to DPC data in detail. The GRASP algorithm can use the temporal and spatial continuity
of pixels, and allow a group of pixels to be inverted at the same time. The multi-pixel retrieval unit
for DPC in the study is shown as **Figure 1**. Each small cube represents a pixel in geographic grids
with a spatial resolution of 0.1°×0.1° (3×3 DPC pixel averaged). This is in accordance with the
MODIS 04_L2 product (~10 km). The projection is determined by the DPC data. Each pixel is
guaranteed to have at least 3 different observation angles. Size of the retrieval unit can be arbitrarily
selected, but limited by the hardware memory. Different colors show the percentage of land or sea,
and usually do not change with time. They need to be clearly defined in GRASP to select different
surface reflectance models. Cloud and no-data pixels need to be removed before the retrieval,
because the cloud flag setting has not been implemented in the current version of code. Finally, this
retrieval unit was applied in the GRASP to calculate the AOD distributions and compared with
AERONET observations.

### 3.4 Settings of Retrieval Parameters

The settings of initial value and spatial-temporal constraint can significantly impact results of



the statistically optimized strategy in the GRASP algorithm (Dubovik et al. 2011). The GRASP
allows different strategies to fit observations. As the cases recorded in the GRASP software, there
are two retrieval schemes. The configurations of the two schemes are different only by settings of
aerosol size distribution in the forward model. One fits the aerosol size distribution with 16 triangle
bins from the range of 0.05 to 15.0 μm, while the other uses 5 lognormal bins at 0.1, 0.1732, 0.3,
1.0, and 2.9 μm, based on pre-calculated optimized kernels of the POLDER-3. The 5 lognormal bins
scheme increases speed by ~9 times (2.5GHz CPU) without any graphical acceleration compared
to the 16 triangle bins scheme, and it has been used to generate the operational PARASOL/GRASP
aerosol products (Chen et al. 2020). In addition, there is a scheme that is being tested called
"GRASP/Model". This fits observational signal by externally mixing several aerosol types with
fixed optical parameters, which is more stable and faster to calculate the AOD.

258        A tolerable absolute error in radiative transfer calculations is set to 0.0005 and the multiple
scattering effects has been considered. Number of atmospheric layers is set to 10 with an exponential
distribution. The input data of the GRASP algorithm was both normalized radiative measurements
at 443, 490, 565, and 670 nm and DOLP of 490 and 670 nm. The initial guess of aerosol and surface
properties are default in the GRASP software. They comply with general principles and are applied
to calculate AOD at a global scale. The Ross-Li's model (Li et al. 2001) and the Cox-Munk model
(Cox and Munk 1954) were used for modeling radiative (non-polarized) reflectance over land and
ocean, respectively, while, the surface polarized reflectance was following the method of Nadal and
Bréon (1999). Among them, the complex refractive index and surface properties are generally
allowed to be fitted as wavelength-dependent parameters in iterations. All constraints on values are
given a default sizeable range, such as the first parameter in the Ross-Li's model allowed to vary
from 0.001 to 1.100. By light scattering calculations (Dubovik et al. 2006), all aerosol microphysical
parameters are converted into optical parameters to participate in radiative simulation. Spatial and
temporal constraints of variabilities of aerosol and surface properties are realized by using Lagrange
multiplier method. More details can be referred to Dubovik et al. (2021). In this study, the
GRASP/Model scheme was used to retrieve AOD from DPC. All calculations of the GRASP relied
on the supercomputing system in the Supercomputing Center of Wuhan University.
**4. Results and Discussions**
4.1 Validation of DPC/GRASP with AERONET

277        As shown in **Figure 2,** the AERONET observations were used as the references to estimate the
performance of AOD retrieval from DPC images based on the GRASP algorithm. Linear regression,
correlation coefficient (R), Normalized Mean Square Error (NMSE), Mean Bias (MB), percentage
falling into Expect Error (EE%), and matching Number (N) were also calculated. Overall, the DPC
GRASP/Model AOD matches the AERONET observations with an R of 0.8590, a MB of 0.0189,
and a NMSE of 0.1432. Nearly 80% of the GRASP/Model AOD retrievals fall within the expect
error bounds, showing a good performance without any quality control. While, the slope of linear
regression was 0.8438, less than 1. This means that under heavy aerosol loading, the DPC/GRASP
may underestimate the AOD. Although the additional radiometric correction factors were applied,
negative drift due to DPC instrument attenuation probably reduces signals from strong reflectance
and thus results lower values of AOD.





288   In order to further study the retrieval performance of GRASP/Model, control the quality of the
retrieval result from DPC data, we calculated the dependences of NMSE with retrieval residuals,
serial length and effective pixel number in retrieval units, and observation geometry, as shown in
**Figure 3**. The retrieval absolute MB showed an obvious increase when the SCA is large than 150°.
Critical observation conditions, such as pixels at the edge of the image, will probably result to a
larger error in both satellite sensor and forward model. By contrast, different viewing angle number
(3-11) have relatively little impact on the retrieval results, that the average absolute MB bias varies
between 0.0395 and 0.0541. The same phenomenon was also found in the **Figure 3c**. With increase
in length of retrieval units, the absolute MB was relatively stable, only fluctuating around 0.047.
This indicated that the fitting scheme for using the external mixing of different aerosol types in this
scheme of the GRASP/Model did not show much dependence of the length of the time series. By
contrast, the absolute MB showed a decrease trend with the number of averaged pixels, from 0.082
to 0.041. It means that the GRASP/Model is relative sensitive to surrounding pixels in the study. In
addition, the spatial-temporal constraints in the retrieval are also affected by Lagrange multipliers,
which can be customized in the configuration file.

303   Fitting residual is an important factor to estimate the quality of retrieval in GRASP. It was
found that the absolute MB showed a slight increase (from 0.047 to 0.063) when the radiative fitting
residuals were larger than 8%. While, the absolute MB had a trend to decrease first and then increase,
with increase in the polarized fitting residuals. Given that the DPC designed uncertainty is about 5%
for radiometric measurements and 0.02 for DOLP, the relatively large absolute MB (0.069) at 0.01
of the polarized fitting residuals is caused by overfitting of GRASP/Model. To summarize, the SCA,
number of averaged pixels, and fitting residuals showed the impacts on DPC GRASP/Model AOD
retrieval in this test. Pixels with SCA > 150, number of averaged pixels < 4, non-polarized fitting
residual < 8%, and 0.01 < polarized fitting residual < 0.08, were removed as the low-quality
retrievals.

313   **Figure 4a** showed the scatterplots and density distributions of DPC/GRASP AOD versus the
AERONET observations after quality control. About a quarter of the points was removed. It was
found that the performance of AOD retrieval from DPC images showed an enhancement. For DPC
GRASP/Model, the R increased from 0.8590 to 0.8982, the EE% increased from 79.58% to 83.16%,
the NMSE decreased from 0.1432 to 0.1008, and the MB decreased from 0.0189 to 0.0176. The
slope of linear regression also showed a slight improvement with the value increasing from 0.8438
to 0.8867. **Figure 4b** displayed the relative frequency of differences between DPC and AEROENT
AOD. The peak values of deviation for DPC GRASP/Model were found at 0.0144, -0.0185, and -
0.0935 when the AOD < 0.2, $0.2 \leq$ AOD < 0.5, and AOD $\geq$ 0.5, respectively. This shows that the
MB drifts from positive to negative as AOD increases.

323   4.2 Evaluation of DPC AOD Performance at a Spatial Scale

324   The DPC AOD retrieved by the GRASP/Model was compared with AERONET observations
at each individual site to show a world-wide retrieval result as **Figure 5**. The R, NMSE, MB, and
EE% were calculated and displayed on sites where the matching number of pixels was larger than
10. In addition to the observation performance of the DPC itself, spatial variations in performances
of AOD retrieval greatly depend on settings of initial parameter and constraint in the GRASP,
whether they are in line with the local aerosol and surface environments. Results showed that the
GRASP/Model achieved a great performance in different regions. The high values of R (> 0.8) were


found in most regions, while the several lower values (~0.6) were mainly observed in North America
and South Africa. The NMSE showed the values of NMSE in most sites were less than 0.1. This
means that ~70% values of AOD retrieval matched the true values very well. In several sites, such
as western United States, the NMSE were larger than 2, revealing that the AOD has a relatively
larger deviation calculated from DPC images based on current parameter setting with the GRASP
algorithm in the regions. The values of AOD were overestimated (~0.05) in the most areas, as shown
in MB of **Figure 5c**. By contrast, the underestimations were found in high aerosol loading regions,
such as South Asia and North Africa, that MB values were between -0.02 and -0.06, in accordance
with the slope of linear regression of less than 1. The EE% showed that over 80% of AOD retrieved
in sites can fall within the expect error. However, an abnormal relatively high EE% (> 60%) from
GRASP/Model was also found in the western United States where the NMSE was large and R was
low. By compared with sites in central Africa, this phenomenon was probably due to the clean air
and extremely low aerosol content there, and thus the NMSE showed relatively larger. It is worth
noting that the parameterization in the GRASP/Model scheme is a globally consistent configuration
in this study and does not consider the characteristics between different regions. This means that it
is possible to achieve better results in local regions by adjusting different parameterizations.
To further estimate the performance of DPC/GRASP AOD, two regions were selected as cases
as shown in **Figure 6**. The MODIS MAIAC, DT, and DB aerosol products were used as comparisons.
It was noted that the DB algorithm was only executed over land in the C6.1 MODIS DB aerosol
products. It was found that the spatial coverage of GRASP/Model AOD from DPC over land was
slightly lower than the MAIAC MODIS aerosol products. In addition to the narrower field of view
and longer re-visit cycle on DPC (MODIS operated in two satellite: Terra and Aqua), the cloud mask
method probably also mis-classified the cloud-free pixels in heavy aerosol loading conditions. This
also partially resulted the underestimation of DPC AOD because the heavy aerosol loading pixels
are removed. Nevertheless, DPC still properly captures the spatial distribution of AOD. The highest
AOD values (> 1.0) in the southern part of China (mainly Guangdong and Guangxi) were caught
by the current retrieval strategy. This is in accordance with the three MODIS products. By contrast,
the AOD found in North China Plain and Centre China by the DPC GRASP/Model (~0.5) were a
little bit lower than MAIAC and DT products (~0.6). However, the DT aerosol products showed
higher AOD in this region, closed to ~1.0. This phenomenon owes to unsuitable aerosol models,
which further results a persistent overestimation in DT algorithm (Che et al. 2019). By the additional
radiometric and polarimetric correction, the DPC GRASP/Model showed good performance over
both Land and Ocean. The high values of AOD in the South China Sea and the estuary of the Yangtze
River can be clearly captured. To summarized, the DPC showed spatial ability of AOD retrieval
based on GRASP algorithm in China region and the similar results have also been reported recently
by using the GRASP/component module (Li et al. 2022).
Another case was selected in Western Europe where the air is clean and aerosol loading is low
(< 0.2) in the most of time around year. As shown in **Figure 6b**, different satellites and aerosol
retrieval methods showed slightly different distributions of AOD. In addition to the different transit
times between DPC and MODIS, this phenomenon is also probably because the aerosol signal is
difficult to separate from the totally satellite observation under low aerosol loading conditions and
thus result relative larger uncertainties of retrieval. From the AOD maps of DPC GRASP/Model,
the relatively high values of AOD (~0.25) were found in Central France, Southern Spain, and
Southern England. While, the MODIS MAIAC showed lower AOD (~0.1) over the mainland and
two points of high AOD (~0.5) were found in Northern coastal areas of Spain and Algeria. By
contrast, the distributions of AOD calculated by DT and DB algorithm were also different from that
calculated by DPC GRASP/Model and MAIAC. The high AOD (~0.4) region appeared in Northern
France, Italy, and Southern England. Compared with single pixel-based retrieval algorithm (such as
DT and DB), the GRASP and MAIAC considered more temporal and spatial information of aerosol
and surface parameters. All of them have been proven to have good performance of AOD retrieval
(Sayer et al. 2014; Lyapustin et al. 2018; Chen et al. 2020; Ou et al. 2021).
### 4.3 Comparison of DPC AOD with MODIS Products at a Temporal Scale
In this section, time-series of AOD were evaluated by compared with MODIS aerosol products
based on the observations of AERONET site. The mean error ratios (MER) were calculated for the
global collocation data set from 23 selected AERONET stations, as shown in **Figure 7**. The MER
compares the mean bias for each satellite aerosol products in a specified period of time to their EE%
(Gupta et al. 2018). Lower absolute value of MER means the smaller actual errors, indicating a good
match with the AERONET. The selected AERONET stations had relatively continuous observations
during the study period to avoid that global validation statistics shift in local emphasis and introduce
temporal variation in the global results (Gupta et al. 2018). From the **Figure 7**, it was found that the
time series of AOD from DPC GRASP/Model had a good matching with the AERONET AOD. The
absolute values of MER were stable and less than ~0.05 after day 65. While the reason of relatively
large negative MER (~0.1) before day 65 is presumed to be low EE%, as the DPC GRASP/Model
would underestimate AOD under heavy aerosol loading conditions. This result is similar to the result
of DT algorithm. Both showed good performances. In addition, the temporal averaged MER showed
that the MODIS DT (0.0230) and DPC GRASP/Model (0.0049) generally overestimated the AOD,
while the MODIS MAIAC (-0.0208) underestimated. By contrast, though the temporal averaged
MER of MODIS DB was closer to 0, this was due to the cancellation between positive and negative
biases. It is worth noting that the same parameter scheme (including start points and constraints)
was applied globally in the GRASP/Model. Therefore, the difference in aerosol optical properties
and spatial-temporal heterogeneity in different regions may be not considered appropriately. The
optimization of the region is expected to improve the inversion effect.
**Figure 8** showed three cases at different underlying surface to display the time series of AOD
retrieved from DPC GRASP/Model on the basis of AERONET observations. The DT AOD was also
compared as a reference, due to its stable performance. It was found that the behavior of AOD from
DPC/GRASP and MODIS DT was generally consistent with AERONET at the three sites. From the
scatterplots, the values of R were 0.983 and 0.928, 0.943 and 0.959, and 0.967 and 0.859 for MODIS
DT and DPC GRASP/Model at Pilar_Cordob, Magurele_Inoe, and FZJ-JOYCE, respectively. The
GRASP/Model AOD retrieved from DPC were slightly higher than the AERONET in the FZJ-
JOYCE site and thus it resulted a relatively lower R. Nevertheless, in general, DPC/GRASP has a
good ability to capture the temporal variation of aerosols.
**Conclusion and Summary**
The DPC/ GaoFen-5 is the first multi-angular polarized sensor launched by China and thus it
has occupied an important position in the development of satellite sensors. In this study, AOD was
retrieved from the DPC images by using the GRASP algorithm and compared with AERONET and



MODIS observations. The main purpose is to evaluate the performance of the DPC to monitor global
aerosols.
On a global basis, a uniform parameterization scheme, which defined the variation ranges and
start values of the optical and microphysical properties (realized by aerosol type) of the aerosol, was
applied in the "Model" module of GRASP. Validations against AERONET showed that the R and
EE% of DPC GRASP/Model were 0.8590 and 79.68%, respectively, in the first attempt. The SCA,
number of averaged pixels in retrieval units, and fitting residual showed an impact on the results of
AOD. A larger number of pixels in retrieval units and a smaller fitting residual can help improve the
quality of retrieval. By quality control (SCA > 150, number of averaged pixels < 4, non-polarized
fitting residual < 8%, and 0.01 < polarized fitting residual < 0.08 removed), the R and EE% of DPC
GRASP/Model improve to 0.8982 and 83.16%, respectively. The corresponding MB and NMSE
decreased from 0.0189 and 0.1432 to 0.0176 and 0.1008, respectively. This indicated that DPC has
a good ability to detect aerosols under this scheme.
In the perspective of spatial scale, the R and EE% of GRASP/Model were larger than 0.9 and
80% respectively in the most AERONET sites. Large NMSE and Low EE were found in low aerosol
loading conditions such as west of the United States. When the actual AOD is small, the retrieval
bias of AOD from satellite observations will be amplified as reflected in NMSE and EE to some
extent. By compared with MODIS aerosol products, the AOD from DPC GRASP/Model showed
good consistency in China, that all regions with high AOD values were detected. Evaluation of the
time-serial AOD showed the performance of DPC GRASP/Model is similar to the MODIS DT and
better than MODIS DB and MAIAC products. Therefore, to summarize, the DPC can capture spatial
and temporal variations in aerosols. The study improves to our understanding of DPC and find a
solution for retrieving AOD based on GRASP algorithm. The continuous development of multi-
angle sensors polarized plays an important role in aerosol monitoring in the future.
**Acknowledgement**
This study was funded by the National Key R&D Program of China (Grant No. 2018YFB0504500),
National Natural Science Foundation of China (Grant No. 41875038, No. 42071348, and No.
42001291), the Key R&D projects in Hubei Province (Grant No. 2021BCA220) and supported by
the LIESMARS Special Research Funding. We are grateful to the Moderate Resolution Imaging
Spectroradiometer (MODIS) Team, the Aerosol Robotic Network (AERONET) Organization and
the GaoFen-5 Directional Polarimetric Camera (DPC) Developed Team for freely distributed their
aerosol products and measurements. The numerical calculations in this paper have been done on the
supercomputing system in the Supercomputing Center of Wuhan University. Finally, we would also
like to thank all reviewers for their constructive and valuable comments.

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

Camera Based on the Rayleigh Scattering over Ocean. *Remote Sensing, 14*

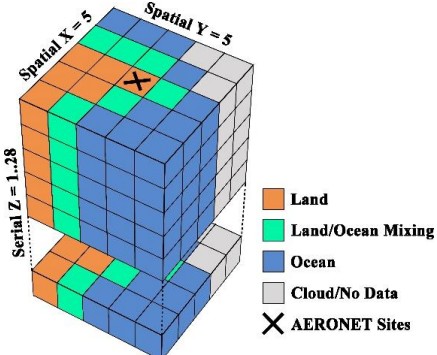


**Figure 1.** Schematic diagram for multi-pixel retrieval unit (5×5×1..28). A maximum of 28 sequences
allowed in each unit is limited by hardware memory.

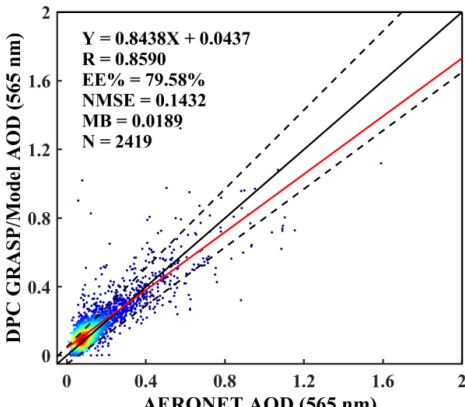


**Figure 2.** Two-dimensional density scatterplot of AOD retrieval from DPC with the GRASP/Model
scheme versus the AERONET observations. The solid black lines are diagonal and the dashed black
lines show the ranges of expect error. The red solid lines represent the linear regression lines.

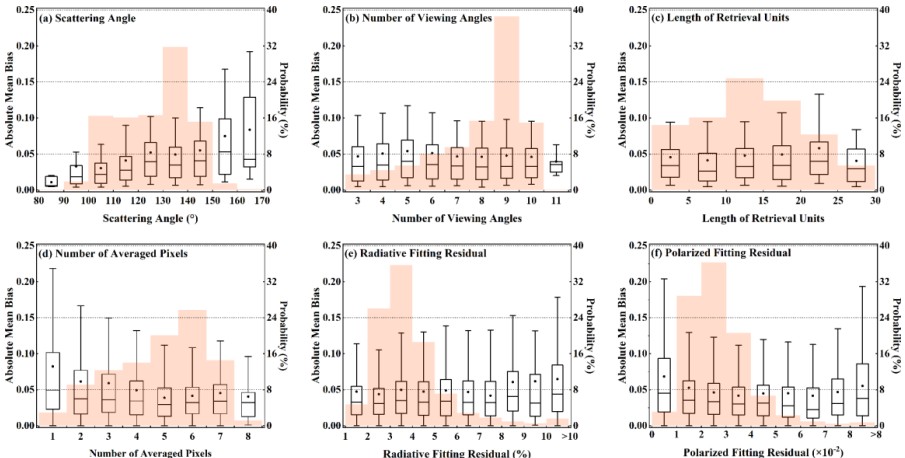

**Figure 3.** Influencing factors of AOD retrieval performance of DPC based on the GRASP/Model:
**(a)** SCA; **(b)** number of viewing angles; **(c)** length of retrieval units; **(d)** number of averaged pixels;
**(e)** non-polarized fitting residual; **(f)** polarized fitting residual. Orange shadows in the background
represents the probability distribution of the samples.

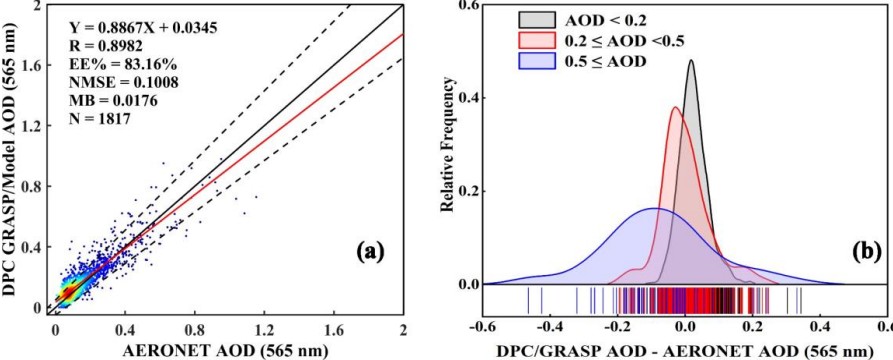

**Figure 4.** Performances of AOD retrieval from DPC data based on the GRASP/Model after quality
control. **(a)** Two-dimensional density scatterplot of AOD retrieval from DPC with the GRASP
algorithm versus the AERONET observations; **(b)** Relative Frequency of AOD differences between
DPC/GRASP and AERONET.





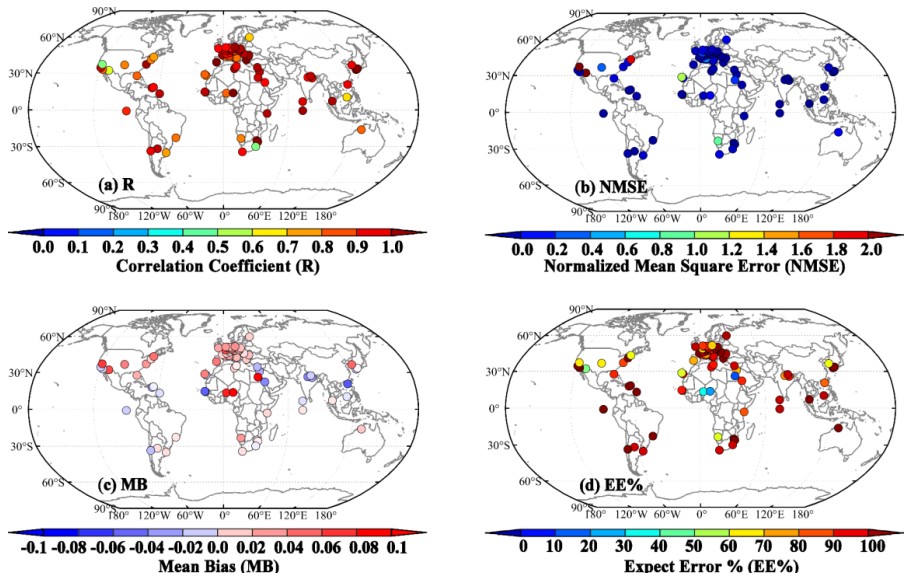

**Figure 5**. Spatial Distributions of **(a)** R, **(b)** NMSE, **(c)** MB, and **(d)** EE% calculated from DPC GRASP/model by compared with AERONET observations. Only sites with more than 10 matching points are included.

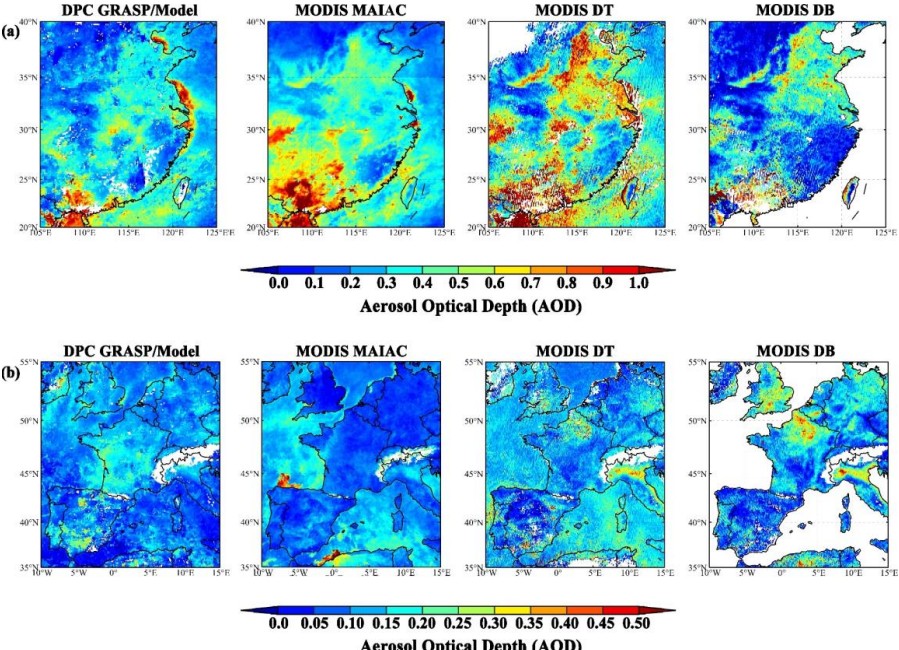

**Figure 6**. Spatial distribution of AOD from DPC GRASP/Model compared with MODIS MAIMC, DT, and DB aerosol products in March, 2020: **(a)** Eastern and Southern China with its adjacent sea areas. The dashed line is part of the Nine-dotted Line; **(b)** Areas of Western Europe including the





Atlantic Ocean and the Mediterranean. The DPC AOD is at 565 nm and the MODIS AOD is at 550
nm.

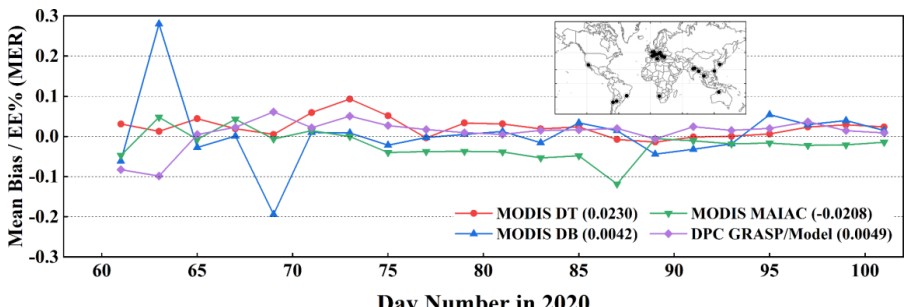


**Figure 7**. Time series of mean error ratios (MER) for the global collocation data set from 23 selected
AERONET stations during March and April of 2020. The number in brackets are temporal averaged
values of MER. The map inset shows the positions of AERONET stations with more details.

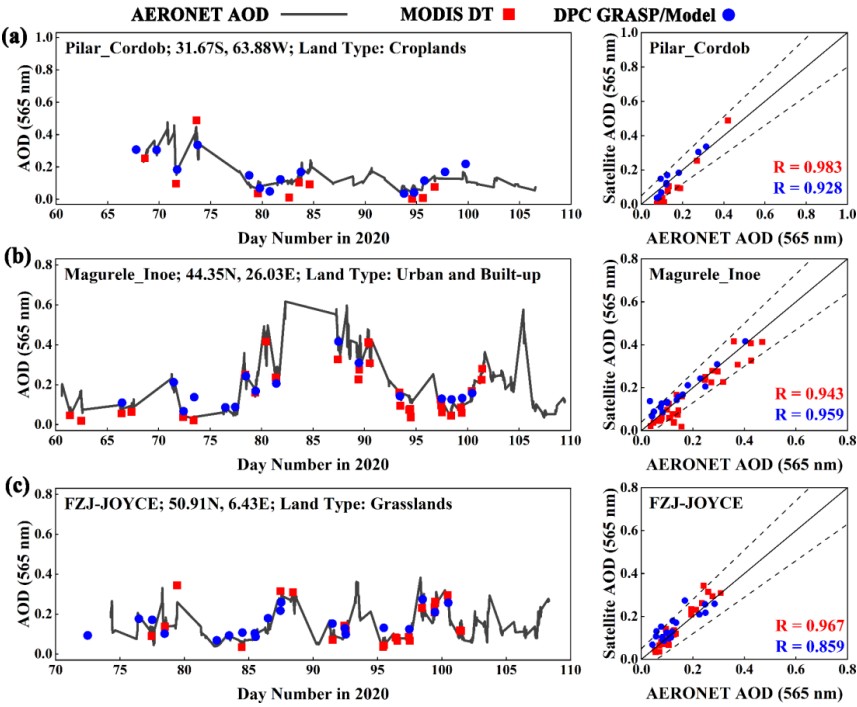


**Figure 8**. Time series of AOD from the DPC GRASP/Model versus the MODIS DT products and
AERONET observations at three sites as cases: **(a)** Pilar_Cordob, **(b)** Magurele_Inoe, and **(c)** FZJ-
JOYCE. The scatterplot shows the relationship between AERONET AOD and satellite AOD.