# Peer review of "Performance Evaluation for Retrieving Aerosol Optical Depth from"

_Atmospheric Measurement Techniques, 2022_

## Referee Comment (RC2)

Review of paper:

**Performance evaluation for retrieving aerosol optical depth from directional polarimetric camera (DPC) based on GRASP algorithm** *by S. Jin et al.*

**Highlights**
- alternate aerosol retrieved by a new satellite sensor with the GRASP algorithm
- investigations of relative relevance of particular sensor data on retrieval accuracy
- comprehensive evaluation

**Concerns**
- still significant differences in spatial distributions
- with all the extra sensor information … no superiority compared to existing retrievals
- missing absorption and size evaluations, hinder meaningful AOD retrieval assessments

**General comments**

A GRASP-based retrieval algorithm is applies to a new Chinese satellite sensor operating since 2018. The Directional Polarimetric Camera (DPC) is a multi-spectral, multi-angle and also polarization sensing instrument to offers a wealth of information about atmosphere and surface. All this information is processed in a statistically optimized GRASP retrieval for a consistent determination of surface and all aerosol properties at cloud free condtions.

Retrieval results for AOD are compared to AERONET local statistics and different MODIS versions and indicate general skill. However, AOD spatial distribution samples still leave many questions open. For a more comprehensive AOD comparison/evaluation – especially for comparisons to other satellite data complementary information on aerosol size (e.g. AOD fine-mode fraction) and absorption e.g. AAOD or even better AAOD attributed to fine and coarse mode) would go a long way.

Otherwise a nice contribution

**Specific comments**

251     are these effective radii (in um)? Since the size-modes are represented by log-normal distributions what are mode(-number) radius und std dev (width information)? For dust regions I would allow another super-large (e.g radius ca 6-9um) dust size, as large size mineral dust, if present, will add significant absorption, which otherwise may be incorrectly attributed to fine-mode aerosol

322     …which is common for aerosol retrieval with most sensors

439     yes as this is kind of a pre-cursur to ESA's upcoming 3MI space sensor

647     show, in addition, the same results side-by side in a log/log scale so info on behavior at low (or most coomon) AOD is better illustrated (the linear fit is less meaningful, as controlled by a few larger values)

651     nice … what about statistics at 1 or 2 (like SLSTR) viewing angles ?

658     the 4b figure is so much better to understand than figure 4a! If there would be similar 4b plots for regions this would be perfect.

668    the comparisons to other satellite data is an eye-opener. There ARE differences that need more attention. It is interesting that for E.Asia MODIS DT greater than MODIs DB, while it is the other way around for western Europe. I also would add MISR data (the are available) for the same region

I attach a seasonal subset of a general (year-independent) MAC reference, which addresses aerosol amount, size and absorption (not just AOD !) for testing satellite retrievals, to identify major retrieval biases (which can be quickly done, if monthly 1x1 averages are provided).

[Figure]

***Figure*** *Seasonal distributions for mid-visible AODf, AAODf, AODc, AAODc of the MAC aerosol climatology. Values to the lower left indicated seasonal global averages.*

---

## Referee Comment (RC3)

[referee-annotated manuscript omitted]

---

## Author Comment (AC1)

**Respond to Reviewer #1**

Dear reviewer, thanks for your comments. We have carefully read your comments, and replied to your comments point by point with corresponding modifications in the manuscript. In the following, your comments are marked in bold italics, our responses are in black, and the modifications in manuscript are shown in blue.

*This study estimates the aerosol detect performance of DPC and verifies the accuracy of AOD retrieval with the GRASP Algorithm. It provides a complete set of DPC data pre-processing flow, and introduces the necessary information of the DPC and GRASP algorithms. In addition to the AERONET, the results of AOD were also compared with various MODIS standard aerosol products at spatial and temporal scales. Generally, the manuscript has been well organized and written. It is worthy for publication after some minor modifications. The comments were as follow. Major Comments:*

Thanks for your comments and recognitions. We have carefully checked your comments and revised the manuscript. The detailed replies are as follows.

*1) The DPC/Gaofen-5 is the first multiple angles and polarization satellite sensor developed by China. Thus, it is recommended to add a brief introduction to the DPC sensor in the Abstract section to help readers quickly understand the sensor.*

Thank you for these suggestions. We added some appropriate additions in the Abstract to introduce the DPC sensors. We believe it is useful for readers. The specific modifications are as follows:

**Line 20-25:** Directional Polarimetric Camera (DPC) is the first generation of multi-angle polarized sensor developed by China. It is onboard GaoFen-5 satellite, running in 705 km sun-synchronous orbit with a 13:30 pm ascending node. The sensor has three polarized channels at 490, 670, and 865 nm and ~9 viewing angles, mainly used for observing aerosols. The spatial resolution is ~ 3.3 km at nadir and global coverage is in ~2 days.

*2) For the Method section, I noticed an additional radiometric correction applied to the DPC data prior to AOD inversion. Is this necessary and does it have a big impact on the results?*

Thanks for your comments. Yes, this step is important to control data quality of DPC. From the report of pervious study [1], the DPC reflectance has a negative bias and it can reach -20% after launch. Therefore, the retrieval of aerosols is difficult to perform without attempting to correct for these large deviations.

[1] Zhu, S., Li, Z., Qie, L., Xu, H., Ge, B., Xie, Y., Qiao, R., Xie, Y., Hong, J., Meng, B., Tu, B., & Chen, F. (2022). In-Flight Relative Radiometric Calibration of a Wide Field of View Directional Polarimetric Camera Based on the Rayleigh Scattering over Ocean. Remote Sensing, 14 (DOI: 10.3390/rs14051211)

*3) As mentioned in Section 3.3, the setting of multi-pixel retrieval unit in the GRASP can help to improve result of AOD inversion. What is the basis for this setting? Does a larger inversion unit mean better inversion results?*

Thanks for your comments. The size of multi-pixel retrieval units can be customized according to the experimental needs. In most cases, the setting of it depends on computer hardware conditions: the larger the retrieval unit, the larger the memory required, and the slower the calculation speed. This multi-pixel concept introduced by Dubovik et al. (2011) allows for benefiting from a priori knowledge on spatial and temporal variability of retrieved parameters and therefor helps to obtain more accurate aerosol retrieval. For example, it is well known that land surface reflectance changes very slowly in time, while aerosol properties have limited spatial variability. Some more discussion can be found in Dubovik et al. (2011, 2021) . As shown in Figure 3 in the manuscript, the increase of ambient pixels and timesteps reduces the inversion bias.

[1] Dubovik, O., Herman, M., Holdak, A., Lapyonok, T., Tanré, D., Deuzé, J.L., Ducos, F., Sinyuk, A., & Lopatin, A. (2011). Statistically optimized inversion algorithm for enhanced retrieval of aerosol properties from spectral multi-angle polarimetric satellite observations. Atmospheric Measurement Techniques, 4, 975-1018, doi:10.5194/amt-4-975-2011

[2] Dubovik, O., Fuertes, D., Litvinov, P., Lopatin, A., Lapyonok, T., Doubovik, I., Xu, F., Ducos, F., Chen, C., Torres, B., Derimian, Y., Li, L., Herreras-Giralda, M., Herrera, M., Karol, Y., Matar, C., Schuster, G.L., Espinosa, R., Puthukkudy, A., Li, Z., Fischer, J., Preusker, R., Cuesta, J., Kreuter, A., Cede, A., Aspetsberger, M., Marth, D., Bindreiter, L., Hangler, A., Lanzinger, V., Holter, C., & Federspiel, C. (2021). A Comprehensive Description of Multi-Term LSM for Applying Multiple a Priori Constraints in Problems of Atmospheric Remote Sensing: GRASP Algorithm, Concept, and Applications. Frontiers in Remote Sensing, 2:706851, doi:10.3389/frsen.2021.706851

*4) In the result of AOD, the large absolute mean bias also appeared when the residual of polarized fitting is small (0.01). The reason of it should be explained in the text.*

Thank you for the comment. The GRASP allows customization of the conditions for stopping iterative fitting and accounts for the noise in the input observations under assumption of that the number of independent observations is significantly larger than the number of retrieved parameters. The situation with very small polarized residuals ($< 0.01$) likely related with the situations when the observations cover rather narrow range of scattering range and large noise is possible. This is only an explanation that response to the abnormal phenomenon.

Another reviewer made similar observations and recommended that we revise the experiment. So, we have fixed the issue after re-performing the experiment.

*Minor Comments:*

*1) Line 25: Abbreviations should be given full names on their first occurrence.*

Thanks for the suggestion. We have fixed this issue now.

*2) Line 29-32: Please rephase the sentence.*

Thanks for the suggestion. This sentence has been rephrased.

**Line 33-36**: Compared with MODIS products, the spatial and temporal variations of aerosol could be caught by the DPC with the GRASP/Models, showing a good performance. However, values of AOD were also underestimated by DPC, probably due to over screening high AOD event by cloud mask.

*3) Line 221: The reference related Fmask is missing.*

Thanks for the suggestion. We have added this reference.

---

## Author Comment (AC2)

**Respond to Reviewer #2**

Dear reviewer, thank you for your useful comments. We have carefully analyzed your comments, and replied to your comments point by point and included corresponding modifications in the manuscript. In the following text, your comments are marked in bold italics, our responses are in black, and the modifications in manuscript are shown in blue.

***Highlights***
***- alternate aerosol retrieved by a new satellite sensor with the GRASP algorithm***
***- investigations of relative relevance of particular sensor data on retrieval accuracy***
***- comprehensive evaluation***

Thank you for the comments. The DPC is the first multi-angle polarized sensor in China and this study is to estimate its performance for aerosol monitor. We hope and believe China's satellite product will be improved with continuous developments and publicly released.

***Concerns***
***- still significant differences in spatial distributions***
***- with all the extra sensor information … no superiority compared to existing retrievals***
***- missing absorption and size evaluations, hinder meaningful AOD retrieval assessments***

Thank you for the comments. We agree with your concerns. In addition to the use of new retrieval methods, the differences in spatial distribution of AOD is probably also related to other factors including different orbitals, narrower swath width [1] (1850 km of DPC, while that of MODIS is ~2300 km), and cloud mask. Since, only a limited set of DPC data (1/3/2020 to 10/4/2020) is used in our application and thus time span is not large. The AERONET retrieval of complex aerosol optical properties (SSA, etc.) use sky-scanning measurement that are less frequent than direct triple observation of the sun (for only AOD). Therefore, only few matches between DPC and AERONET could be obtained in the range of 25 km/30 min. More importantly, the current study is mainly a preliminary attempt to analyze the DPC data that were strongly compromised by 20% bias… Therefore, our analysis is mainly dedicated to base aerosol parameters such as total AOD, since the retrieval of more detailed properties generally requires very high quality observations. Nonetheless to address the comments we provide the **Figure** below showing preliminary comparisons between daily average SSA and fine-mode AOD from DPC GRASP/Models and AERONET data, as a brief assessment of aerosol absorption and size information. Overall, the result are somewhat comparable to the POLDER-3 GRASP/Models results reported by Chen et al 2020 [2]. Also, as discussed by Chen et al. (2020) it can be noted, that the GRASP/Models was the approach that provided the least accurate retrieval of details aerosol parameters (SSA and AODF) while was the best for AOD.

We hope this illustration can, at least, partially address your concerns. We will make continuous efforts in the future to data quality and usability of Chinese sensors.

[Figure]

**Figure.** Comparisons of daily SSA and fine-mode AOD from DPC GRASP/Models and AERONET.

[1] Li, Z., Hou, W., Hong, J., Zheng, F., Luo, D., Wang, J., Gu, X., & Qiao, Y. (2018). Directional Polarimetric Camera (DPC): Monitoring aerosol spectral optical properties over land from satellite observation. Journal of Quantitative Spectroscopy and Radiative Transfer, 218, 21-37, doi:10.1016/j.jqsrt.2018.07.003.

[2] Chen, C., Dubovik, O., Fuertes, D., Litvinov, P., Lapyonok, T., Lopatin, A., Ducos, F., Derimian, Y., Herman, M., Tanré, D., Remer, L.A., Lyapustin, A., Sayer, A.M., Levy, R.C., Hsu, N.C., Descloitres, J., Li, L., Torres, B., Karol, Y., Herrera, M., Herreras, M., Aspetsberger, M., Wanzenboeck, M., Bindreiter, L., Marth, D., Hangler, A., & Federspiel, C. (2020). Validation of GRASP algorithm product from POLDER/PARASOL data and assessment of multi-angular polarimetry potential for aerosol monitoring. Earth System Science Data, 12, 3573-3620, doi:10.5194/essd-12-3573-2020

*General comments*

*A GRASP-based retrieval algorithm is applies to a new Chinese satellite sensor operating since 2018. The Directional Polarimetric Camera (DPC) is a multi-spectral, multi-angle and also polarization sensing instrument to offers a wealth of information about atmosphere and surface. All this information is processed in a statistically optimized GRASP retrieval for a consistent determination of surface and all aerosol properties at cloud free condtions. Retrieval results for AOD are compared to AERONET local statistics and different MODIS versions and indicate general skill. However, AOD spatial distribution samples still leave many questions open. For a more comprehensive AOD comparison/evaluation – especially for comparisons to other satellite data complementary information on aerosol size (e.g. AOD finemode fraction) and absorption e.g. AAOD or even better AAOD attributed to fine and coarse mode) would go a long way. Otherwise, a nice contribution*

Thanks for the thoughtful comment. GRASP is the most notable algorithm to this problem, and it has good prospects and has shown convincing results in POLDER-3. Therefore, we decided to follow it. Nonetheless, we agree that obtaining good retrievals of detailed aerosol optical and microphysical need more efforts especially taking into account the current known issues with DPC data availability and quality (20% bias, etc.). Thanks for your understanding, we will continue to work on this in the future.

*Specific comments*

*251 are these effective radii (in um)? Since the size-modes are represented by log-normal*

*distributions what are mode(-number) radius and std dev (width information)? For dust regions I would allow another super-large (e.g radius ca 6-9um) dust size, as large size mineral dust, if present, will add significant absorption, which otherwise may be incorrectly attributed to finemode aerosol*

Thanks for your comments. These are the mode radius rather than effective radius. In the module of GRASP/Optimized, the aerosol size distribution was fit by 5 lognormal bins with mode radii of 0.1, 0.1732, 0.3, 1.0, and 2.9 μm, and mode standard deviations of 0.35, 0.35 ,0.35, 0.5, and 0.5, correspondingly. It should be noted that the contribution of the supper large particles is accounted in those bins. As for the retrieval of details of size concentrations for those super large particles, such retrieval would over ambitious, since the information content in reflected radiation measured from space is rather limited for this ambitious task. This aspect was discussed by Dubovik et al. (2011) and therefore number of parameters of describing the size distribution was significantly reduced for satellite retrieval compare to AERONET retrievals (see also Dubovik et al., 2021).

[1] Dubovik, O., Herman, M., Holdak, A., Lapyonok, T., Tanré, D., Deuzé, J.L., Ducos, F., Sinyuk, A., & Lopatin, A. (2011). Statistically optimized inversion algorithm for enhanced retrieval of aerosol properties from spectral multi-angle polarimetric satellite observations. Atmospheric Measurement Techniques, 4, 975-1018, doi:10.5194/amt-4-975-2011

*322 …which is common for aerosol retrieval with most sensors*

Thanks for the thoughtful comment. Would you mean general principle for aerosol retrieval? In the forward model of GRASP, surface reflectance is modeled by Ross-Li BRDF and Nadal BPDF, and aerosol is modeled by considering complex optical and microphysical properties. There are no fixed assumptions. To avoid misleading readers, we also revised the text.

**Line 268-273**
The initial guess of aerosol and surface properties are default in the GRASP software. They are applied to calculate AOD at a global scale. The Ross-Li's model (Li et al. 2001) and the Cox-Munk model (Cox and Munk 1954) were used for modeling radiative (non-polarized) reflectance over land and ocean, respectively, while, the surface polarized reflectance followed the method of Nadal and Bréon (1999). More detailed description can be found in Dubovik et al. (2011),

*439 yes as this is kind of a pre-cursur to ESA's upcoming 3MI space sensor*

Thanks for your comments. Yes, I learned about the 3MI through the official website. This is a wonderful project that we are looking forward to.

*647 show, in addition, the same results side-by side in a log/log scale so info on behavior at low (or most coomon) AOD is better illustrated (the linear fit is less meaningful, as controlled by a few larger values)*

Thank you for the suggestions. We agree that presentation in log scale provides some additional

inside on the error distribution, therefore we have added such figures. The added figure in the manuscript as shown below. At the same time, we are not sure if this representation will be fully convincing for the readers as we have rarely seen the log scale coordinates before, therefore we keep original figures too.

[Figure]

**Figure 2. (a)** Density scatterplot of AOD retrievals from DPC with the GRASP/Models scheme versus the AERONET observations with a linear coordinate system. **(b)** The density scatterplot with a logarithmic coordinate system. The solid black line is the one to one and the dashed black lines show the ranges of Expected Error. The red solid lines represent the linear regression line; **(c)** Box plots show changes of differences between DPC GRASP/Models and AEROENT with AOD increasing. Diamond marks and curves represent distributions of sample and normal distribution fitting lines, respectively.

*651 nice … what about statistics at 1 or 2 (like SLSTR) viewing angles?*

Thank you for the suggestions. In this study, we performed cloud screening separately for different angles, since they were not registered at the same time exactly. We further removed those data that only had 1-2 observation angles, in order to prevent the influence of cloud edges. From the principle of GRASP, if new observational errors will not be introduced with increasing number of angles, it is beneficial to the retrieval of complex properties of aerosols. Therefore, it is expected that, unilaterally, the retrieval performance of the retrieval using multiple angles should have benefits compare to use of 1-2 angles is weaker than that of, but it should be enough for only AOD retrieval in GRASP.

*658 the 4b figure is so much better to understand than figure 4a! If there would be similar 4b plots for regions this would be perfect.*

Thank you for the suggestions. We have added AOD difference figures between DPC GRASP/Models and MODIS DT, DB, and MAIAC products, as shown below. We hope this can add some quantitative statistics to our study.

[Figure]

**Figure 7**. Distribution Density of AOD differences between DPC GRASP/Models and MODIS DT, DB, and MAIAC products at: **(a-c)** Eastern and Southern China with its adjacent sea areas; **(d-e)** Areas of Western Europe including the Atlantic Ocean and the Mediterranean. It is noted that the MODIS DB product only releases terrestrial AOD data.

*668 the comparisons to other satellite data is an eye-opener. There ARE differences that need more attention. It is interesting that for E.Asia MODIS DT greater than MODIs DB, while it is the other way around for western Europe. I also would add MISR data (the are available) for the same region I attach a seasonal subset of a general (year-independent) MAC reference, which addresses aerosol amount, size and absorption (not just AOD !) for testing satellite retrievals, to identify major retrieval biases (which can be quickly done, if monthly 1x1 averages are provided).*

Thanks for your comments. In terms of spatial analysis, the differences may come from the shorter time span, which is affected by the amount of approved data. Such short-term and large-scale analyses are relatively rare, so we did not want to expand the scope of this study to draw an inappropriate conclusion.

The MISR is a multi-angular detector that can obtain aerosol-scale information. Whereas, our initial investigation indicates that there is a clear systematic underestimation of its AOD in aerosol high loading regions, thus have not considered it for now. In addition, we noticed that a new generation research algorithm optimized for MISR has been proposed (https://amt.copernicus.org/preprints/amt-2022-95/). That study well solves the problem by using a priori MODIS/MAIAC BRDF dataset for high aerosol loading cases, and we are also very much looking forward to its new aerosol product.

Finally, for further complex aerosol optical properties (size and absorption), we calculated them with DPC GRASP/Models as shown following. It must be stated that retrieval of aerosol complex properties depends on to high-quality observations. Since the DPC has large bias, the figure below is just a preliminary result for only reviewing. I used a similar color to your figure for better contrast,

and the 10AAOD means ten times of AAOD (10AAOD = 10 × AAOD). I hope this can partially resolve your confusion. The current study is only a preliminary attempt. We will make continuous efforts in the future to data quality and usability of Chinese sensors.

[Figure]

**Figure.** Distributions of AOD, 10AAOD, Fine-mode AOD, and Coarse-mode AOD retrieved from DPC GRASP/Models during 01/03/2020 to 20/03/2020. I guess the 10AAOD in your figure means ten times of value of AAOD.

---

## Author Comment (AC3)

**Respond to Reviewer #3**

Dear reviewer, thank you for your useful comments. We have carefully analyzed your comments, and replied to your comments point by point and included corresponding modifications in the manuscript. In the following text, your comments are marked in bold italics, our responses are in black, and the modifications in manuscript are shown in blue.

*The paper outlines an application of the GRASP algorithm to retrieval of aerosol optical depth from observations of the DPC multiangle polarimeter. Results are validated against the AERONET sun-photometer network, from which quality control metrics are devised, and a qualitative comparison is made to three MODIS aerosol products. The performance appears to be consistent with other remotely sensed aerosol products, which is impressive for a relatively new satellite and research team.*
*While I found the presentation generally good, I was disappointed by the meagre details provided by this manuscript. It would be impossible to replicate the method from this paper alone and the quantitative evaluation covers only AERONET. However, the authors have done a better job than many published works so I expect to see this work in print after some misunderstandings are corrected.*

Thank you for the thoughtful comments. The manuscript can provide information about the application of GRASP/Models on the DPC data. Although the DPC data are non-public, the GRASP software is open source code that available at GRASP-OPEN web site (https://www.grasp-sas.com/). The details of GRASP/Models approach are described in several publication, e.g., see Lopatin et al. (2021) and Dubovik et al. (2021)
Thanks for pointing out the mistakes in our manuscript. This will be very helpful for us to improve the quality of the manuscript.

[1] Lopatin, A., Dubovik, O., Fuertes, D., Stenchikov, G., Lapyonok, T., Veselovskii, I., Wienhold, F.G., Shevchenko, I., Hu, Q., & Parajuli, S. (2021). Synergy processing of diverse ground-based remote sensing and in situ data using the GRASP algorithm: applications to radiometer, lidar and radiosonde observations. Atmos. Meas. Tech., 14, 2575-2614, doi:10.5194/amt-14-2575-2021
[2] Dubovik, O., Fuertes, D., Litvinov, P., Lopatin, A., Lapyonok, T., Doubovik, I., Xu, F., Ducos, F., Chen, C., Torres, B., Derimian, Y., Li, L., Herreras-Giralda, M., Herrera, M., Karol, Y., Matar, C., Schuster, G.L., Espinosa, R., Puthukkudy, A., Li, Z., Fischer, J., Preusker, R., Cuesta, J., Kreuter, A., Cede, A., Aspetsberger, M., Marth, D., Bindreiter, L., Hangler, A., Lanzinger, V., Holter, C., & Federspiel, C. (2021). A Comprehensive Description of Multi-Term LSM for Applying Multiple a Priori Constraints in Problems of Atmospheric Remote Sensing: GRASP Algorithm, Concept, and Applications. Frontiers in Remote Sensing, 2:706851, doi:10.3389/frsen.2021.706851

*In the following, S means section and L means line number.*
- *At a glance, there is a substantial overlap between this paper and Li et al. 2022 as both apply GRASP to DPC. They evaluate different measurands and only share two authors, but my experience is that the AOD products discussed here are a by-product of the aerosol type*

*products discussed there. The manuscript before me certainly provides additional information and I am not questioning the logic in publishing the projects separately. However, there must be more clarification of the relationship between the teams, either acknowledging how their work has complemented each other (e.g. I would hope that their determination of aerosol type provided the inputs to this method) or explaining why it was necessary to make separate implementations of the same code (to assist future GRASP users in determining which to use)?*

Thank you for the comment. Are you referring to the article by Lei Li et al [1] published on the *Atmospheric Research* in March 2022? Our study is completely independent of them. This study was independently initiated by Wuhan University and completed with the help from GRASP team (Dr. Chen and Prof. Dubovik) and DPC team (Dr. Hong). Authors are selected in accordance with actual contributions. In fact, Li's article is more like a case study of pollution events in eastern China without any actual quantitative assessment, neither AOD nor type products. By contrast, our study is a quantitative assessment of DPC/GRASP AOD on a global scale according to AERONET. It's not a same story for our perspective. In addition, Dr. Hong is one of the heads of DPC, mainly responsible for the calibration work, and therefore, we didn't rush our experiments until we fully figured out the DPC calibration performance. Whereas, in the meantime, a study of a comprehensive assessment of radiation measurements of DPC was published on *Remote Sensing* in March 2022 [2]. It depicted large uncertainties of the DPC signals (can be up to 20%) and provided a correction method. We don't know whether Li's study have taken these into account. Therefore, although I think his research is good, we did not benefit from research extensively in our manuscript.

There is a significant difference of Lei Li et al [1] and current studies. Lei Li et al [1] implemented GRASP/Components approach while in present studies we used GRASP/Models approach. While GRASP/Components approach retrieved more parameters and provides more extensive set of aerosol parameters, GRASP/Models approach uses more constraints on retrieved aerosol and shows very stable and convincing performance for total AOD that is superior over other GRASP approaches (e.g. see Chen et al. (2020) and Dubovik et al. (2021)). Therefore, knowing the issues with high DPC uncertainties we have chosen GRASP/Models approach and focused our study on the analysis of AOD retrievals.

[1] Li, L., Che, H., Zhang, X., Chen, C., Chen, X., Gui, K., Liang, Y., Wang, F., Derimian, Y., Fuertes, D., Dubovik, O., Zheng, Y., Zhang, L., Guo, B., Wang, Y., & Zhang, X. (2022). A satellite-measured view of aerosol component content and optical property in a haze-polluted case over North China Plain. Atmospheric Research, 266, 105958 (DOI: 10.1016/j.atmosres.2021.105958)

[2] Zhu, S., Li, Z., Qie, L., Xu, H., Ge, B., Xie, Y., Qiao, R., Xie, Y., Hong, J., Meng, B., Tu, B., & Chen, F. (2022). In-Flight Relative Radiometric Calibration of a Wide Field of View Directional Polarimetric Camera Based on the Rayleigh Scattering over Ocean. Remote Sensing, 14 (DOI: 10.3390/rs14051211)

[3] Levy, R.C., Remer, L.A., & Dubovik, O. (2007). Global aerosol optical properties and application to Moderate Resolution Imaging Spectroradiometer aerosol retrieval over land. Journal of Geophysical Research Atmospheres, 112 (DOI: 10.1029/2006JD007815)

[4] Chen, C., Dubovik, O., Fuertes, D., Litvinov, P., Lapyonok, T., Lopatin, A., Ducos, F., Derimian, Y., Herman, M., Tanré, D., Remer, L.A., Lyapustin, A., Sayer, A.M., Levy, R.C., Hsu, N.C., Descloitres, J., Li, L., Torres, B.,

Karol, Y., Herrera, M., Herreras, M., Aspetsberger, M., Wanzenboeck, M., Bindreiter, L., Marth, D., Hangler, A., & Federspiel, C. (2020). Validation of GRASP algorithm product from POLDER/PARASOL data and assessment of multi-angular polarimetry potential for aerosol monitoring. Earth System Science Data, 12, 3573-3620 (DOI: 10.5194/essd-12-3573-2020)

[5] Dubovik, O., Fuertes, D., Litvinov, P., Lopatin, A., Lapyonok, T., Doubovik, I., Xu, F., Ducos, F., Chen, C., Torres, B., Derimian, Y., Li, L., Herreras-Giralda, M., Herrera, M., Karol, Y., Matar, C., Schuster, G.L., Espinosa, R., Puthukkudy, A., Li, Z., Fischer, J., Preusker, R., Cuesta, J., Kreuter, A., Cede, A., Aspetsberger, M., Marth, D., Bindreiter, L., Hangler, A., Lanzinger, V., Holter, C., & Federspiel, C. (2021). A Comprehensive Description of Multi-Term LSM for Applying Multiple a Priori Constraints in Problems of Atmospheric Remote Sensing: GRASP Algorithm, Concept, and Applications. Frontiers in Remote Sensing, 2:706851, doi:10.3389/frsen.2021.706851

- *Throughout the paper the authors report Expect Error (EE%), being the number of retrieved values falling within some range of the validation value, and comment positively when these increases. Putting aside the fact that the authors never define the term, nor state the envelope they use, this misunderstands the meaning of an error envelope. The MODIS error envelope is an estimate of a normally distributed error derived from comparison to validation data. As such, only 68% of data should fall within the error envelope (see "one sigma confidence interval"). Achieving a higher EE% does not mean the data is "better", merely that the EE overestimated the uncertainty in the circumstances considered. Ideally, the authors would estimate their own error envelope, which would presumably be narrower than that of MODIS. At a minimum, though, the authors must revise the language to express that the ideal EE% is 68%. (Also, Expected Error would be more grammatically correct.)*

Thank you for the comment. According to your correction, we are fully aware that existing expression of EE% in the manuscript can cause some problems and misunderstandings. Therefore, we modified it and added an explanation based on your suggestion. In addition, we would like to clarify that the relatively high EE% in DPC/GRASP AOD is only used to show the good performance of DPC in our manuscript. We do not expect (or do not think) that the EE% can be used alone to evaluate the performance of AOD products. The specific modification is as follow:

Line 288-296:
Linear regression, correlation coefficient (R), Root Mean Square Error (RMSE), Mean Bias (MB), percentage falling into Expected Error **(EE%, ±(0.05+0.15\*AOD))**, and matching Number (N) were also calculated. Among them, the EE% is selected in accordance with the MODIS error envelop and **the ideal EE% is ~68% under assumption of normal distribution within one sigma confidence interval**. Therefore, the EE can be used to estimate approximately the accuracy MODIS AOD. Overall, the DPC GRASP/Models AOD matches the AERONET observations with an R of 0.8511, a MB of 0.0256, and a RMSE of 0.0842, showing good performance without any quality control. Nearly 80% of the GRASP/Models AOD retrievals fall within the Expected Error bounds, **revealing that the error envelop of DPC is probably narrower than that of MODIS**.

- *While being clear that I don't expect the authors to change anything in the paper as they follow common and widespread practice, I will point out that the evaluation provided does not actually assess the accuracy of their retrievals. It assesses the accuracy of 30min/25km*

*averages of their retrievals. Thus, the variability shown is a lower bound for the method's performance. This is clearly demonstrated in Fig.3(d), where accuracy improves as more observations are aggregated.*

Thank you for the thoughtful comments. Yes, as you point out, the accuracy improves with more observations aggregated. But here is one more thing to clarify. The "number of averaged pixels" is also an indicator for GRASP, because the GRASP takes into account the surrounding pixels in the retrieval of aerosols. The finding that "accuracy improves as more observations are aggregated" also suggests that multi-pixel retrieval have better performance than sing-pixel retrieval especially for GRASP.

● *The MODIS Dark Target, Deep Blue and MAIAC products are widely used, so I understand why the authors compare to them. But why do they not compare to MISR (or another polarimeter), which would provide a like-for-like comparison to another multiangle retrieval and demonstrate the relative merits of the GRASP method?*

Thank you for the comment. Our main purpose is to study (or assess) the DPC rather than the GRASP, while the GRASP is a well-developed and flexible algorithm that has been applied on many instruments. I know there are several multi-angle or polarization payloads in orbit, such as the SGLI/GCOM-C (Japan), but it is difficult to access to their products. For MISR, our initial investigation indicates that there is a clear systematic underestimation of its AOD in aerosol high loading regions, thus we do not use them either. In addition, we noticed that a new generation algorithm optimized for MISR has been proposed (https://amt.copernicus.org/preprints/amt-2022-95/). That study well solves the problem by using MODIS/MAIAC BRDF dataset, and we also look forward to its new aerosol product.

● *I am more disappointed that, given the number of satellites it has been applied to, there is no comparison to another implementation of GRASP. That would provide valuable insight into the relative performance of the DPC sensor independent of retrieval method and assumptions.*

Thank you for the comment. I am sorry for making you feel that. The main purpose is to study DPC in our research. To be honest, we have tested several implementations of GRASP on DPC observations, and the GRASP/Models module was select after balancing the performance and calculation speed. For instance, for an implementation that uses 16 bins to fit the aerosol volume size distribution without any optimization (as case recorded in GRASP software), a unit with 300 pixels can take more than an hour to compute (2.5 GHz Xeon CPU). Therefore, only the results from GRASP/Models approach are present in our manuscript.

● *S2.3) Please be more precise as to the data used. Do you use every AERONET site in the record or do you exclude some? Do you report every collocation or do you exclude some? Do you use Level 1.5 or Level 2.0 as using both would be extremely foolish?*

Thank you for the comment. In the original results, we used all the sites that matched DPC data,

including Level 1.5 and Level 2.0. Whereas, to avoid being foolish :), we revised the study and only used Level 2.0 data in the revised manuscript. All 178 sites with available Level 2.0 AOD during the study period participated in the validation and we also re-evaluated the results. The specific modification is as follow:

**Line 156-157:**

The AOD data used for validation were acquired from all **178 AERONET sites** with available Level 2.0 AOD products during the study period, which have been cloud-screened and quality controlled.

- *L215-7) I do not know what you mean by 'absolute value of average relative deviations'. An equation would be clearer.*

Thank you for the comment. We have added two equations for clarification. When the value of whiteness test is greater than 0.7, the pixel is considered to be cloudy.

**Line 219-223:**

$$MeanVis = (Band_1 + Band_2 + Band_3)/3 \qquad (2)$$
$$Whiteness\ Test = \sum_{i=1}^{3}|(Band_1 - MeanVis)/MeanVis| > 0.7 \qquad (3)$$

Where, $Band_1$, $Band_2$, and $Band_3$ are reflectance in red, green, and blue bands received by satellite at top of the atmosphere, respectively. Corresponding to the DPC, they are 490, 565, and 670 nm, respectively.

- *S3.3) You don't appear to do any cloud filtering before averaging. Why not, given how common such approaches are in other aerosol retrievals?*

Thank you for the comment. We have done the cloud screening before averaging. The cloud screening is done after radiometric calibration and it precedes all other retrieval operations.

- *L236) The text states that the retrieval unit is 3x3 but Fig.1 shows a 5x5 unit. I appreciate that the larger cube helps illustrate the inclusion of different surfaces types in a single retrieval but please clarify what, precisely, is being done.*

Thank you for the comment. There seems to be misunderstanding. The 3x3 averaged was performed on the raw DPC data, to reduce the amount of data and improve the signal-to-noise ratio of the aerosol signal and make the spatial resolution close to that of the MODIS product. This means that in retrieval, 9 DPC pixels were averaged into one pixel in a retrieval unit. In other words, this 3x3 represents a down-sampling process. Whereas, the 5x5xNT (number of time layers) in the Fig. 1 means the scale of retrieval unit which was used for GRASP multi-pixel retrieval in our study. For example, assuming that the AOD of an AERONET site is to be calculated, then we can take it as a center and select 5x5 (total 25) pixels around this site to put into the GRASP model. In GRASP retrieval (or iteration), the loss function needs to take a global minimum for these a group of pixels. In this way, the multi-pixel retrieval is achieved, instead of the well-known pixel-by-pixel (single-pixel) retrieval. In our validation activity, we used the 5x5 window for aggregation, while in the POLDER-3 validation [1], it used 3x3 for land and 9x9 for ocean, while for the retrieval it used

2x2xNT. This means that the scale of retrieval unit can be changed as needed. A large scale of retrieval unit usually reveals that the more surrounding pixel information is considered, but the more memory is needed.

[1] Chen, C., Dubovik, O., Fuertes, D., Litvinov, P., Lapyonok, T., Lopatin, A., Ducos, F., Derimian, Y., Herman, M., Tanré, D., Remer, L.A., Lyapustin, A., Sayer, A.M., Levy, R.C., Hsu, N.C., Descloitres, J., Li, L., Torres, B., Karol, Y., Herrera, M., Herreras, M., Aspetsberger, M., Wanzenboeck, M., Bindreiter, L., Marth, D., Hangler, A., & Federspiel, C. (2020). Validation of GRASP algorithm product from POLDER/PARASOL data and assessment of multi-angular polarimetry potential for aerosol monitoring. Earth System Science Data, 12, 3573-3620 (DOI: 10.5194/essd-12-3573-2020)

- *Fig.2) I strongly agree with Dr. Kinne that this figure should be shown on a logarithmic scale. doi:10.5194/acp-19-15023-2019 provides compelling evidence that linear averaging of AOD provides misleading conclusions.*

Thank you for the comment. I agree with you and so I have added the density scatterplots and showed them on logarithmic scale. As Prof. Sayer points out in 10.5194/acp-19-15023-2019 that AOD is often distributed close to log-normally on large scales, the usage of the logarithmic coordinate system can display the scatter plot of AOD more clearly. Meanwhile, we keep the plot using linear scale, which also provide useful information especially for high AOD cases.

[Figure]

**Figure 2. (a)** Density scatterplot of AOD retrievals from DPC with the GRASP/Models scheme versus the AERONET observations with a linear coordinate system. **(b)** The density scatterplot with a logarithmic coordinate system. The solid black line is the one to one and the dashed black lines show the ranges of Expected Error. The red solid lines represent the linear regression line; **(c)** Box plots show changes of differences between DPC GRASP/Models and AEROENT with AOD increasing. Diamond marks and curves represent distributions of sample and normal distribution fitting lines, respectively.

- *L284) I disagree that Fig.2 shows that you underestimate AOD in high loading circumstances. I read that plot as showing an underestimate of AOD in typical circumstances (as the red blob is above the black line; see also the grey curve of Fig.4b). Your best-fit line has a gradient less than one largely because it's going through the peak of the distribution around 0.1 and towards the handful of points around 1.2.*

Thank you for the comment. Yes, linear regression line may lead a wrong conclusion as you point

out. Therefore, to further explain, we added a figure to show how the bias changes with the AOD increasing, as **Figure 2(c)** mentioned above. Despite the limited sample (12 points), it clearly shows that the DPC AOD is underestimated when the AERONET AOD is greater than 0.8. We have made corresponding changes in the text.

**Line 297-300**

This means that under heavy aerosol loading, the DPC/GRASP probably underestimate the AOD. More details are presented in **Figure 2c**. It is found the lower slope of linear regression is mainly controlled by several points which have larger AOD (> 0.8). By contrast, when AOD is less than 0.8, the retrieval is stable.

**Line 329-333**

**Figure 4c** displayed the change of differences between DPC and AEROENT AOD. The underestimations when AOD > 0.8 were not found to be restrained by the quality control. A possible reason is that an overly restrictive cloud mask can remove aerosol pixels during heavy pollution. In addition, the negative drift after the launch of the DPC may also be the reason, if it is not fully corrected.

- *L286) There are numerous sources of error in any AOD retrieval and I would be surprised if the aging of the detector was the primary one.*

Thank you for the comments. Our research is applying existing methods to a new sensor. It means we are more inclined to find the reason from the DPC hardware when analyzing the error, because we have had a preliminary understanding of the performance of GRASP/Models from other previous studies. In addition, more importantly, the DPC has a severe negative drift in the radiation calibration results after launch. This is why we used additional correction coefficients in the study. If we do not use the additional correction coefficients, the most immediate result is a significant underestimation of AOD. Here we refer a retrieval case by Prof. Zhengqiang Li's team, as following **Figure** (http://www.sonet.ac.cn/yjdt/html/?200.html=) [1]. Therefore, when there is an underestimation in our results, our inference is the issue of DPC radiometric calibration. The possible cause of this negative drift is the aging of the instrument (probably the battery). While, in order not to cause misunderstandings by readers, we have also revised this sentence.

[Figure]

**Figure. (a)** AOD retrievals **without** the additional correction coefficients; **(b)** AOD retrievals **with** the additional correction coefficients.

[1] Zhu, S., Li, Z., Qie, L., Xu, H., Ge, B., Xie, Y., Qiao, R., Xie, Y., Hong, J., Meng, B., Tu, B., & Chen, F. (2022). In-Flight Relative Radiometric Calibration of a Wide Field of View Directional Polarimetric Camera Based on the Rayleigh Scattering over Ocean. Remote Sensing, 14 (DOI: 10.3390/rs14051211)

**Line 329-333**
Figure 4c displayed the change of differences between DPC and AEROENT AOD. The underestimations when AOD > 0.8 were not found to be restrained by the quality control. A possible reason is that an overly restrictive cloud mask can remove aerosol pixels during heavy pollution. In addition, the negative drift after the launch of the DPC may also be the reason, if it is not fully corrected.

- *L297) On L255, you said that the method of external mixtures was under testing, which I took to mean "an experimental mode that will eventually be available". The text here implies that that is the mode you used. Please clarify what was done.*

Thank you for the comments. In this study, the GRASP/Models is applied on the DPC data and it is a first try based on the implementation of POLDER GRASP/Models application. The current version of GRASP/Models already has good performance and has been tested on POLDER-3 [1] and DPC (our study). However, it still could improve in the setting of coarse particle optical properties, definition of models etc. So, the sentence there means that the GRASP-SAS will continue to improving the GRASP/Models approach, and the current version in our study is not final.

[1] Chen, C., Dubovik, O., Fuertes, D., Litvinov, P., Lapyonok, T., Lopatin, A., Ducos, F., Derimian, Y., Herman, M., Tanré, D., Remer, L.A., Lyapustin, A., Sayer, A.M., Levy, R.C., Hsu, N.C., Descloitres, J., Li, L., Torres, B., Karol, Y., Herrera, M., Herreras, M., Aspetsberger, M., Wanzenboeck, M., Bindreiter, L., Marth, D., Hangler, A., & Federspiel, C. (2020). Validation of GRASP algorithm product from POLDER/PARASOL data and assessment of multi-angular polarimetry potential for aerosol monitoring. Earth System Science Data, 12, 3573-3620 (DOI: 10.5194/essd-12-3573-2020)

- *L308) The retrieval residuals should conform to some distribution, such that very small values are not unexpected. Why is the polarized component different, requiring the exclusion of small residuals? My gut instinct would be a systematic bias in the observations or something about the representation of Rayleigh scattering in the forward model.*

Thank you for the comments. The GRASP allows customization of the conditions for stopping iterative fitting and accounts for the noise in the input observations under assumption of that the number of independent observations is significantly larger than the number of retrieved parameters. The situation with very small polarized residuals (< 0.01) likely related with the situations when the observations cover rather narrow range of scattering range and large noise is possible. This is only an explanation that response to the abnormal phenomenon.
After revision (replacing combination of Level 1.5 and Level 2.0 with Level 2.0 for AERONET), the abnormal phenomenon (AOD results with larger biases but the polarized fitting residuals were lower) has been alleviated. Thus, we removed this requirement to perform validation (polarized

fitting residual < 0.01). In addition, distributions of fitting residual are shown as following **Figure**.

[Figure]

**Figure. (a)** Distribution of radiative fitting residual; **(b)** Distribution of polarized fitting residual. The curves are log-normally fitting lines.

**Line 322-324:**

Retrieval is considered low quality if any of the following conditions are met: 1) Pixels with SCA > 150; 2) number of averaged pixels < 4; 3) length of timesteps < 5; 4) non-polarized fitting residual > 8%; 5) polarized fitting residual > 0.06.

- *Fig.6) This is an entirely qualitative comparison. There's nothing necessarily wrong with that, but I feel a page of description is inappropriate for two hand-picked scenes that cannot represented general performance. Perhaps the figure could appear at the start of the section as an illustration of the approach? Also, I feel the authors have failed to mention the most important feature of this diagram: their method exhibits minimal land-sea contrast compared to others. This is a long-standing advantage of GRASP retrievals and limitation of other methods that is not widely acknowledged.*

Thank you for the comments. To add a quantitative evaluation, differences between DPC GRASP and other MODIS aerosol products were calculated as **Figure 7** in the revised manuscript (also showing as follow). From this figure, the DPC GRASP/Models AOD is still different from other MODIS products that cannot be ignored. The most obvious feature is the underestimation of AOD when the aerosol loading is heavy.

Also, we agree with you that the GRASP has the smallest land-sea contrast and we added some sentences to describe this feature in the revised manuscript.

**Line 386-389**

Compared with single pixel-based retrieval algorithm (such as DT and DB), the GRASP and MAIAC considered more temporal and spatial information of aerosol and surface parameters. And benefit from the consistency of all assumptions (regarding aerosol and a priori constrains), the DPC GRASP exhibits minimal land-sea contrast.

[Figure]

**Figure 7**. Distribution Density of AOD differences between DPC GRASP/Models and MODIS DT, DB, and MAIAC products at: **(a-c)** Eastern and Southern China with its adjacent sea areas; **(d-f)** Areas of Western Europe including the Atlantic Ocean and the Mediterranean. It is noted that the MODIS DB product only releases terrestrial AOD data.

- *Fig.7) While I appreciate an example of the performance over time, I do not like the units chosen (mean error ratio). As the text describes, MER can decrease both because a method has large, but complementary, errors; because the error envelope increases; or because different time steps present more/less difficult retrieval conditions. Further, L387 is not strictly correct as a lower MER can be achieved by a lower Error Envelope, which would happen if one method retrieves larger values than the other. More generally, I'm curious why the authors use normalised mean square error throughout the paper (without specifying by what the error is normalised; different communities would expect the retrieved value or the EE) rather than the more common root-mean square error?*

Thank you for the comment. Initially, the usage of normalized mean square error (NMSE) rather than the more common root-mean square error (RMSE) is in order to reflect the DPC retrieval performance when the AOD is small. But we found that a better approach is to use a logarithmic scale, as you suggested above. So, in the revised version, we changed the NMSE back to RMSE and re-drew the **Figure 8** as following. From this figure, the DPC GRASP/Models AOD shows a good performance with lower average daily RMSE. This is also in accordance with the higher EE%, which reveals that the error envelop of DPC GRASP/Models is probably narrower than that of MODIS. The corresponding parts of the text have also been revised, as follows:

**Line 407-419**
From the **Figure 8**, it was found that the time series of AOD from DPC GRASP/Models had a good matching with the AERONET AOD. The values of RMSE were ~0.06 and stable before 87[th] day. While the reason of relatively large RMSE (~0.12) around 90[th] day is presumed to be heavy aerosol loading conditions, as the DPC GRASP/Models would underestimate AOD under this situation. The

similar temporary rapid increases in RMSE were also found in MODIS products, such as the 80[th] day of the DT, the 85[th] day of the DB, and 98[th] day of the MAIAC. This reflects the time instability of algorithms. In addition, the lowest daily averaged RMSE was found in DPC GRASP/Models with value of 0.0663, and then MODIS DT (0.0863) and MODIS DB (0.0913). The low RMSE of DPC may be due to it ignoring some high value AODs. It is worth noting that the same parameter scheme (including start points and constraints) was applied globally in the GRASP/Models. Therefore, the difference in aerosol optical properties and spatial-temporal heterogeneity in different regions may be not considered appropriately.

[Figure]

**Figure 8.** Time series of daily RMSE for the selected AERONET stations during March and April of 2020. The number in brackets are averaged values of daily RMSE.

- *Some more minor comments:*
-
- *L42) D'Almeida 1991 is a strange reference here, considering it's a microbiology paper.*

Thank you for the comment. I guess there were some problems inserting the reference, but now we have fixed it.

- *L126) By "normalized radiation", do you mean "reflectance"?*

Thank you for the comment.. Yes, the "normalized radiation" is "reflectance" received by satellite sensor (at top of the atmosphere).

- *L147) Please be specific what is meant by "highest quality" as different fields mean different things by it.*

Thanks for pointing out this. We have revised this section and specified the data selection, as following.

**Line 147-150**
The corrected AOD (quality flag = 3) on land and average AOD (quality flag = 1,2,3) on the ocean are selected in the DT products. The best estimated AOD (quality flag = 2,3) is selected in the DB products. The best quality AOD (QA AOD = 0000) is selected in the MAIAC products.

- *L156) These are common collocation criteria, and I am not asking you to change anything*

*here, but you may find it beneficial to read the series of papers Nick Schutgens has published on the best strategy to collocate different aerosol datasets, such as doi:10.5194/acp-20-12431-2020, 10.5194/acp-2015-973 and 10.5194/acp-16-1065-2016.*

Thanks for these instructive comments and information. It's very useful and brings new ideas of collocation and present the validation results, such as Taylor diagram.

- *L187) I disagree. It is entirely feasible to create a look-up-table-based method that integrates different instruments as, if you have an module that perform a calculation, it is possible to build a look-up table from it. The advantages of GRASP lie in its detailed radiative transfer simulations and multipixel approach. If you replace 'traditional look-up table-based methods' with 'most popular retrieval methods' I no longer have a problem with the sentence.*

Thank you for the comment. We have revised this sentence as your suggestion.

**Line 189-191**
This avoids that the most popular look-up table-based methods are difficult to apply to each other, due to the limitations of different sensor channel and characteristic.

- *L230) How often does DOLP>1 happen? Unless it's extremely rare, this filtering feels like it would introduce a low bias into that value.*

Thank you for the comment. The DOLP means ratio of linearly polarized light to total light. When the DOLP is 0, it indicates that the light is unpolarized (such as sunlight), when the DOLP is 0 to 1, it indicates that the light is partially linearly polarized, and when the DOLP is 1, it indicates that the light is fully linearly polarized. Therefore, the DOLP must be less than 1 theocratically. It is also rare that the DOLP is greater than 1 on the DPC, which may be caused by the calibration error of the Stokes parameters.

- *L259-60) Does 'exponential distribution' mean that your vertical levels are spaced exponentially, such that they are roughly equally spaced in pressure?*

Thank you for the comment. The 'exponential distribution' here is relative to several other common vertical distributions of aerosols, such as Gaussian or single layer distribution. It is usually used as a priori assumption on the vertical height of the aerosol, when retrieving aerosols [1]. Because GRASP allows retrieval of parameters related to the vertical height of aerosols, and so this sentence is used to illustrate what vertical assumptions we used. So, in the AOD retrieval, this is not related to the pressure.

[1] Wu, Y., de Graaf, M., & Menenti, M. (2017). The impact of aerosol vertical distribution on aerosol optical depth retrieval using CALIPSO and MODIS data: Case study over dust and smoke regions. Journal of Geophysical Research: Atmospheres, 122, 8801-8815, doi:10.1002/2016jd026355

- *L262) "General principles" is far too generic a description. Either remove it because the text that follows elaborates or explain what you mean.*

Thank you for the comment. We have revised this sentence as you marked in the text.

- *Fig.3) Please reproduce this figure so the text is a similar size to that in the figure's caption.*

Thank you for the comment.. We have re-draw this figure and increase the font size.

- *L296) I think it would be clearer to refer to 'timesteps' rather than 'retrieval units'.*

Thank you for the comment.. We revised and used the "timesteps" instead of the "the length of retrieval units" to express the number of observations in the time dimension.

- *L311) Also on L425. I think you've gotten the direction of the inequality wrong for the residuals, as it currently implies you remove high scattering angle and middline polarized residual. I also think you mean 'or' rather than 'and' as very few points will satisfy all of those conditions simultaneously.*

Thanks for pointing out mistakes. This probably came from editing. We've fixed it now.

**Line 322-324**
Retrieval is considered low quality if any of the following conditions are met: 1) Pixels with SCA > 150; 2) number of averaged pixels < 4; 3) non-polarized fitting residual > 8%; 4) polarized fitting residual > 0.06.

- *L414) It will certainly occupy an important position in China. The impact on the rest of the world will depend on the availability of the data.*

Thank you for the comment. We hope and believe that the disclosure and release system of China's satellite data will be improved.

- *L671) While appreciating that the authors likely have little control over it, I see no reason for the nine-dotted line to appear in this plot as it is a political, rather than physical, boundary.*

Thank you for the comment. We have deleted the related description.

- *I attach an annotated PDF with typographic corrections I hope will be of use. They are largely verb tenses and use of 'the', which I certainly couldn't do accurately in another language. Red lines indicate text to delete while yellow highlight is for word replacement or insertion. As these were done by hand on a tablet, they do not cover precisely the words affected and all capitalisation in my comments should be ignored. Also, many of the citations either lack or incorrectly state the page number of the paper. If the authors have the time, it*

*would be a massive improvement to include the DOIs of papers as this simplifies finding a paper.*

Dear reviewer, we are grateful that you can make such careful revisions, which we have rarely received before. Based on your suggestions, we have revised the text one by one, and added page numbers and DOI information for the references. It is no doubt that your comments are very important to us. Thanks for your review.

---

## Editor Decision (ED1)

Thanks for the nice response to my review.

Given the limited data (not even a year) it is understood that evaluations are somewhat limited. Although the general approach is promising, the admitted 20% bias indicates the need for significant improvements (maybe that can also be stated in the abstract).

The offered comparison of retrieval SSA and fine-mode AOD to Aeronet are quite interesting.
The SSA comparison indicates that retrieval AOD are likely too low (in case of absorbing aerosol).
This also can explain that AODf is mainly too low in case for major (absorbing) wildfire/pollution cases.

On size, AERONET size-distributions cover radii up to 15um, although with limitations of degradation for concentrations of radii of above 5um. Mode radii are given with respect to number of volume – which is why I prefer eff.radii. When the mode radii you listed with respect to number than Rmode 2.3 is close to reff 5um … then no complaints, but if R-mode is with respect to volume (reff is then smaller) then an important big (and solar absorbing) dust size is missed in the retrieval.

Figure 2 is now much better. Retrieval overestimations al low AOD, as in most retrievals, is confirmed (what happended in case of negative AOD … or is this not an option?)

Figure 7 shows that high AOD cases are missed (in part as they are also probably more absorbing than the retrieval allows, as the SSA comparison to AERONET demonstrates).

The comparisons between AOD, 10*AAOD, AODf and AODc are quite interesting (thanks for using a similar color scale) especially when comparing to the relevant DJF season of MAC. However, it looks like that AOD > AODf +AODc … why? Your AAOD seems rather low (compared to MAC). And retrievals over NH land (probably due to snow cover) are largely missing… so having a complete year for evaluation would be more insightful.

Do not work with research algorithms from MISR, these are just as the name said short lived usually with limited coverage … better work with operational products

I like to bias presentation of the bias distribution for the different AOD ranges.

---

## Author Response (AR2)

**Round 2:Respond to Reviewer #2**

Dear reviewer, thanks for your comments. We have carefully read your comments, and replied to your comments point by point with corresponding modifications in the manuscript. It is no doubt that your comments are very important to us. In the following, your comments are marked in bold italics, our responses are in black, and the modifications in manuscript are shown in blue.

***Thanks for the nice response to my review.***

***Given the limited data (not even a year) it is understood that evaluations are somewhat limited. Although the general approach is promising, the admitted 20% bias indicates the need for significant improvements (maybe that can also be stated in the abstract).***

Dear reviewer, thanks for your comments. Yes, this bias was a big problem in our research. But considering that the calibration bias of DPC is the main conclusion of another published paper, we do not emphasize this in our manuscript. But in the error analysis, we must point out the influence of this deviation.

***The offered comparison of retrieval SSA and fine-mode AOD to Aeronet are quite interesting. The SSA comparison indicates that retrieval AOD are likely too low (in case of absorbing aerosol). This also can explain that AODf is mainly too low in case for major (absorbing) wildfire/pollution cases.***

Dear reviewer, thanks for your comments. Yes, it is not excluded that this is the reason for the underestimation of AOD under heavy aerosol loading conditions. The retrieval of aerosol absorption and scattering abilities have always been a challenge, and in any case, from the results, the SSA of GRASP/Models are not optimal among different GRASP implementations.

***On size, AERONET size-distributions cover radii up to 15um, although with limitations of degradation for concentrations of radii of above 5um. Mode radii are given with respect to number of volume which is why I prefer eff.radii. When the mode radii you listed with respect to number than Rmode 2.3 is close to reff 5um then no complaints, but if R-mode is with respect to volume (reff is then smaller) then an important big (and solar absorbing) dust size is missed in the retrieval.***

Dear reviewer, thanks for your comments. Reviewer 3 is also interested in this question. But in the current version of GRASP, the development team hopes to retrieve aerosol parameters using consistent assumptions and constraints in the global. Therefore, a specific aerosol type which only appear in a unique environment have not been considered. Of course, region-specific optimizations should not be excluded either, and we will continue to consider this issue in future research.

*Figure 2 is now much better. Retrieval overestimations at low AOD, as in most retrievals, is confirmed (what happended in case of negative AOD or is this not an option?)*

Dear reviewer, thanks for your comments. Yes, the negative AOD is not an option in the GRASP. In iterations, all parameters must obey strict physical bounds in the GRASP. Currently, the slightly overestimations of AOD under low aerosol loading conditions is ubiquitous in GRASP, and similar phenomenon also occurs in results of POLDER-3/GRASP.

*Figure 7 shows that high AOD cases are missed (in part as they are also probably more absorbing than the retrieval allows, as the SSA comparison to AERONET demonstrates).*

Dear reviewer, thanks for your comments. Yes, we agree with you too, and we will look into this further in future research.

*The comparisons between AOD, 10\*AAOD, AODf and AODc are quite interesting (thanks for using a similar color scale) especially when comparing to the relevant DJF season of MAC. However, it looks like that AOD > AODf +AODc why? Your AAOD seems rather low (compared to MAC). And retrievals over NH land (probably due to snow cover) are largely missing so having a complete year for evaluation would be more insightful.*

Dear reviewer, thanks for your comments. In that figure, the study period is from 2020.03.01 to 2020.03.20, instead of DJF (do this mean December, January, and February?). In the GRASP, the AOD = AODf + AODc. The reason why AOD is looks like > AODf + AODc may be that we are not using a continuous color scale. As mentioned above, since the GRASP/Models results in a higher SSA, it can be expected that the AAOD will be much lower. This is indeed a problem, but the spatial distribution of AAOD is in line with the trend.
We removed data from high latitudes (> 60) for higher calculation speed and the usage of these data is not high as you said they are affected by snow cover. Yes, for a more insightful assessment, we also plan/want to use a complete year of data in future studies.

*Do not work with research algorithms from MISR, these are just as the name said short lived usually with limited coverage better work with operational products.*

Dear reviewer, thanks for your suggestions. Yes, we are only interested in the problems about MISR products revealed by this MISR research algorithm.

*I like to bias presentation of the bias distribution for the different AOD ranges.*

Dear reviewer, thanks for your comments. Yes, we display the bias distribution of different between DPC/GRASPs and MODIS products as shown in Figure 7.

**Round 2:Respond to Reviewer #3**

Dear reviewer, thanks for your comments. We have carefully read your comments, and replied to your comments point by point with corresponding modifications in the manuscript. It is no doubt that your comments are very important to us. In the following, your comments are marked in bold italics, our responses are in black, and the modifications in manuscript are shown in blue.

*I thank the authors for their revisions to the manuscript, which I feel improved it sufficiently to warrant publication. I am particularly pleased with the additional figures. I have a few comments on the reply for the author's to consider in future,*
*- In your reply to Reviewer 2 about L251, you say that "It should be noted that the contributiuon of the super large particles is accounted in those bins", referring to the five lognormal bins of GRASP. While you are correct that the existing bin scheme provides very large particles from the tails of those distributions, this will be correlated to the loading of small particles (i.e. to get lots of large particles requires also having more 2.9um particles). Coarse and fine mode aerosols have different sources and sinks, such that an ideal bin model would have a separate mode for the coarse mode that can be adjusted independently.*

Thank you for these suggestions. In the current version of GRASP, it hopes to retrieve aerosols with a consistent assumption and constraints in the global. Therefore, a specific aerosol type which only appear in a unique environment have not been considered. Of course, region-specific optimizations should not be excluded either, and we will continue to consider this issue in future research.

*- I am intrigued by your finding that MISR systematically underestimates AOD, as my experience with the v23 data has been quite good. This may simply be a cultural difference, but if I'd found a problem in someone else's data, that comparison would have been front-and-centre in my paper. Finally, it should be noted that the MODIS products are not perfect and differing from MODIS does not necessarily mean one is wrong. For example, there is an ongoing discussion about the AOD of remote, clean air.*

Thank you for your comments. Yes, in China, we are more concerned with high aerosol loading conditions because air pollution is still a big problem. Under the low aerosol loading conditions, the performance of MISR is very well.

*- My sincere apologies for misrepresenting your relationship to Li et al. 2022. I did try to check the affiliations, but it can be difficult from Europe to search for details of researchers at Chinese institutions.*

Thank you for the understanding and recognition!

*- When I asked for a comparison to another implementation of GRASP, I wasn't thinking of different GRASP versions applied to DPC. I was thinking of comparison to existing products processed with GRASP, such as POLDER (https://www.grasp-open.com/products/polder-data-release/). By using an existing algorithm like GRASP, your team is well positioned to untangle the errors caused by the algorithm and the errors caused by the instrument.*

Thank you for your comments. Yes, I agree with you, but the problem is that DPC was launched in 2017, whereas, the POLDER-3 have stopped working after 2013.

*- I wish you the best in dealing with the negative drift of DPC. It sounds challenging.*

Thank you for your wishes! We will continue to work on the availability of Chinese satellite data.

*- To be less flippant, I should explain that Levels 1.5 and 2.0 of AERONET are different filterings of the same underlying data, with the latter being more stringent in the removal of possible cloud contamination and applying more nuanced calibration methods. Using both was "foolish" because the dataset would contain duplicate observations. Apologies if this was not clear from AERONET's documentation.*

Thank you for your comments. Yes, we understand this deeply and have made revisions.

*- At the end of section 3.2, you filter out "obvious noise" with DOLP > 1. There may be value in evaluating how this filtering biases your products. While it is true that DOLP > 1 is physically impossible, it can be a valid observations when working with noisy radiances from separate sensors. For example, the co-polar channel could experience an unusually negative random fluctuation, pushing that signal below its dark current, at the same time as the cross-polar one experiences an unusually positive one. An equivalent problem occurs in lidar analysis, where it was found that removing unphysical observations introduced a positive bias into the final products as the filtering had artificially truncated the otherwise symmetrical distribution of random errors.*

Thank you for your comments and experience, it taught me a lot. However, as far as I know, the case of DOLP>1 cannot input and handle in a general forward model. In the future, we will consider and deal with such issues more carefully.

*Some further technical corrections, using line numbers from the document with tracked changes:*
*Figs.2-4) There are horizontal dashed lines at semi-random levels (i.e. at 0.5 and 0.75 in 2(c) but at -0.75, -0.5, 0, 0.75 in Fig.4(c). There's nothing wrong with having a grid but try to be consistent between plots.*

Thank you for your comments. This is mainly caused by the compression of pictures in the word software, and now we have fixed this problem.

Below we show the specific language modifications:

*L25) coverage is ~2 days*

The spatial resolution is ~ 3.3 km at nadir and glob**al coverage is**  **~2 days.**

*L35) From most AERONET sites*

From the most  AERONET sites, the R and EE% were larger than ~0.9 and ~80%.

*L40) ability of the DPC*

The above findings validated the ability of **the** DPC sensor to monitor aerosols.

*L92) with a relatively high spatial resolution*

with **a** relatively **higher** spatial resolution of 3.3 km, that can observe Earth from ~9 different angles.

*L97) There is a space before the period here.*

The multi-angular polarized sensor can provide many more observations for the same pixel in an aerosol parameter **retrieval.**

*L102) the surface accounts*

A well-known advantage is that the polarized light from the surface  accounts for a small part of the total polarized light

*L132) Capitalise Earth.*

that can observe  Earth from ~9 different angles in a local time of ~13:30 PM

*L147) valuable Earth observations*

The Moderate-resolution Imaging Spectroradiometer (MODIS) has been in service for over two decades, providing valuable Earth observations.

*L173) to average satellite data*

a common approach is followed to average satellite data within ±30 min and a circle of 0.25° (~25 km) radius

*L191) in the construction of the modelled reflectance*

surface properties (Bidirectional Reflectance/Polarization Distribution Function, BR/PDF, etc.) **in the construction of the modelled reflectance**.

*L212) the [I,Q,U]^T represent*

where, the $[I, Q, U]^T$  represent the radiative and polarized radiances

*L219-20) "Cloudy pixels are the main factor" or "Cloud contaimination is the main factor"*

**Cloudy pixels are the main factor impacting aerosol retrieval,** because they will block the signal from aerosol due to high reflectance, large coverage, and relatively high vertical position

*L247) to generate a cloud mask*

This feature has been used to **a** generate cloud mask product for both POLDER and DPC sensors

*L271) For instance, the GRASP*

For instance, the GRASP software gives two retrieval schemes for POLDER observations.

*L280-1) "The GRASP/Models approach assumes an external mixture of several" or "The GRASP/Models approach assumes externally mixed aerosols, which"*

**The** GRASP/Models approach assumes **an external mixture of several aerosol types** with fixed optical parameters

*L298) using the Lagrange multiplier*

Spatial and temporal constraints of variabilities of aerosol and surface properties are realized  using Lagrange multiplier method.

*L317) probably underestimates the AOD*

This means that under heavy aerosol loading, the DPC/GRASP probably underestimate**s** the AOD.

*L346) Delete "While" as this statement doesn't preceed a sentence.*

 **The** absolute MB had a trend to decrease first and then increase, with increase in the polarized fitting residuals.

*L362) You say, "Figure 4b displayed the changes." Usually, one refers to things with a document in the present tense (e.g. "displays the changes"). You aren't strictly wrong here - the figures did show those things in the past - but it sounds weird. This also occurs at L436, 475.*

**Figure 4c displayed**s the change**s** of differences between DPC and AEROENT AOD.
**Figure 7 showed**s density distributions of difference between DPC and MODIS products in ranges of AOD
**Figure 9 showed**s three cases at different underlying surface to display the time series of AOD retrieved from DPC GRASP/Models on the basis of AERONET observations

*L379) while the lower values*

The high values of R (> 0.8) were found in most regions, while the  lower values (~0.6) were mainly observed in North America.

*L385) in most areas*

From the MB of **Figure 5c**, the values of AOD were overestimated (~0.04) in  most areas.

*L404) resulted in the underestimation*

This also partially resulted **in** the underestimation of DPC AOD because the heavy aerosol loading pixels are removed.

*L411) This phenonmenon is caused by unsuitable aerosol models, which further results in a persistent overestimation in the DT algorithm*

**This phenomenon is caused by unsuitable aerosol models, which further results in a persistent overestimation in the DT algorithm**

*L418-9) loading is low most of the year*

Another case was selected in Western Europe where the air is clean and aerosol loading is low (< 0.2)  most of the year.

*L438-9) regions. A common pattern is seen in all sub-plots*

 A common pattern  is seen in all sub-plots, namely that the differences were nearly normally distributed

*L439) Either put "and" or a comma after "distributed".*

namely that the differences were nearly normally distributed**,** centered on the 0 under low aerosol

loading conditions (AOD ≤ 0.2).

***L454) period to avoid how global validation statistics shift with the spatial distribution of observations***

The AERONET stations had relatively continuous observations during the study **period to avoid how global validation statistics shift with the spatial distribution of observations**

***L489) "The main purpose was to evaluate" to be consistent with the tense of the previous sentence. Also, in the conclusions we're talking about things that have been done rather than things that are happening now so the past tense is appropriate.***

The main purpose **was** to evaluate the performance of the DPC to monitor global aerosols.

***L506) respectively at most AERONET sites***

In the perspective of spatial scale, the R and EE% of GRASP/Models were larger than 0.9 and 80% respectively at  most AERONET sites.

***L512) overstrict cloud masking***

However, the values of AOD were underestimated by DPC, probably due to overstrict cloud **masking**

***L515-6) The study improves our understanding of DPC and finds a solution***

The study improves to our understanding of DPC and find**s** a solution for retrieving AOD based on GRASP algorithm.

***L676) The page number for Lui 2022 is 106121.***

Liu, B., Ma, X., Ma, Y., Li, H., Jin, S., Fan, R., & Gong, W. (2022). The relationship between atmospheric boundary layer and temperature inversion layer and their aerosol capture capabilities. Atmospheric Research, 271, **106121**, doi:10.1016/j.atmosres.2022.106121

***L688) The page number for Martins 2002 is MOD4 (GRL had a period of weird numbering.)***

Martins, J.V., Tanré, D., Remer, L., Kaufman, Y., Mattoo, S., & Levy, R. (2002). MODIS Cloud screening for remote sensing of aerosols over oceans using spatial variability. Geophysical Research Letters, 29, **MOD4**, doi:10.1029/2001GL013252